# Embedding optimization reveals long-lasting history dependence in neural spiking activity

**Lucas Rudelt**[1]*, **Daniel González Marx**[1], **Michael Wibral**[2], **Viola Priesemann**[1,3]*

**1** Max Planck Institute for Dynamics and Self-Organization, Göttingen, Germany, **2** Campus Institute for Dynamics of Biological Networks, University of Göttingen, Göttingen, Germany, **3** Bernstein Center for Computational Neuroscience, Göttingen, Germany

* lucas.rudelt@ds.mpg.de (LR); viola.priesemann@ds.mpg.de (VP)

**Data Availability Statement:** The data underlying the results presented in the study are available online—from the CRCNS (crcns.org) data repository: http://crcns.org/data-sets/hc/hc-2—

## Abstract

Information processing can leave distinct footprints on the statistics of neural spiking. For example, efficient coding minimizes the statistical dependencies on the spiking history, while temporal integration of information may require the maintenance of information over different timescales. To investigate these footprints, we developed a novel approach to quantify history dependence within the spiking of a single neuron, using the mutual information between the entire past and current spiking. This measure captures how much past information is necessary to predict current spiking. In contrast, classical time-lagged measures of temporal dependence like the autocorrelation capture how long—potentially redundant—past information can still be read out. Strikingly, we find for model neurons that our method disentangles the *strength* and *timescale* of history dependence, whereas the two are mixed in classical approaches. When applying the method to experimental data, which are necessarily of limited size, a reliable estimation of mutual information is only possible for a coarse temporal binning of past spiking, a so-called past embedding. To still account for the vastly different spiking statistics and potentially long history dependence of living neurons, we developed an embedding-optimization approach that does not only vary the number and size, but also an exponential stretching of past bins. For extra-cellular spike recordings, we found that the strength and timescale of history dependence indeed can vary independently across experimental preparations. While hippocampus indicated strong and long history dependence, in visual cortex it was weak and short, while in vitro the history dependence was strong but short. This work enables an information-theoretic characterization of history dependence in recorded spike trains, which captures a footprint of information processing that is beyond time-lagged measures of temporal dependence. To facilitate the application of the method, we provide practical guidelines and a toolbox.

## Author summary

Even with exciting advances in recording techniques of neural spiking activity, experiments only provide a comparably short glimpse into the activity of only a tiny subset of all neurons. How can we learn from these experiments about the organization of information

from the Mendeley Data repository: https://data.mendeley.com/datasets/4ztc7yxngf/1—from the Dryad database: https://datadryad.org/stash/dataset/doi:10.5061/dryad.1f1rc—from the Janelia figshare repository: https://janelia.figshare.com/articles/dataset/Eight-probe_Neuropixels_recordings_during_spontaneous_behaviors/7739750 All code for Python3 that was used to analyze the data and to generate the figures is available online at https://github.com/Priesemann-Group/historydependence.

**Funding:** All authors received support from the Max-Planck-Society, https://www.mpg.de/de. L.R. was supported by the Deutsche Forschungsgemeinschaft (DFG, German Research Foundation) as part of the SPP 2205 - project number 430157073. L.R. acknowledges funding by SMARTSTART, the joint training program in computational neuroscience by the VolkswagenStiftung and the Bernstein Network, https://www.smartstart-compneuro.de/. M. W. is employed at the Campus Institute for Dynamics of Biological Networks funded by the VolkswagenStiftung, https://www.volkswagenstiftung.de. The funders had no role in study design, data collection and analysis, decision to publish, or preparation of the manuscript.

**Competing interests:** The authors have declared that no competing interests exist.

processing in the brain? To that end, we exploit that different properties of information processing leave distinct footprints on the firing statistics of individual spiking neurons. In our work, we focus on a particular statistical footprint: We quantify how much a single neuron's spiking depends on its own preceding activity, which we call history dependence. By quantifying history dependence in neural spike recordings, one can, in turn, infer some of the properties of information processing. Because recording lengths are limited in practice, a direct estimation of history dependence from experiments is challenging. The embedding optimization approach that we present in this paper aims at extracting a maximum of history dependence within the limits set by a reliable estimation. The approach is highly adaptive and thereby enables a meaningful comparison of history dependence between neurons with vastly different spiking statistics, which we exemplify on a diversity of spike recordings. In conjunction with recent, highly parallel spike recording techniques, the approach could yield valuable insights on how hierarchical processing is organized in the brain.

## Introduction

How is information processing organized in the brain, and what are the principles that govern neural coding? Fortunately, footprints of different information processing and neural coding strategies can be found in the firing statistics of individual neurons, and in particular in the history dependence, the statistical dependence of a single neuron's spiking on its preceding activity.

In classical, noise-less efficient coding, history dependence should be low to minimize redundancy and optimize efficiency of neural information transmission [1–3]. In contrast, in the presence of noise, history dependence and thus redundancy could be higher to increase the signal-to-noise ratio for a robust code [4]. Moreover, history dependence can be harnessed for active information storage, i.e. maintaining past input information to combine it with present input for temporal processing [5–7] and associative learning [8]. In addition to its magnitude, the timescale of history dependence provides an important footprint of processing at different processing stages in the brain [9–11]. This is because higher-level processing requires integrating information on longer timescales than lower-level processing [12]. Therefore, history dependence in neural spiking should reach further into the past for neurons involved in higher-level processing [9, 13]. Quantifying history dependence and its timescale could probe these different footprints and thus yield valuable insights on how neural coding and information processing is organized in the brain.

Often, history dependence is characterized by how much spiking is correlated with spiking with a certain time lag [14, 15]. From the decay time of this lagged correlation, one obtains an intrinsic timescale of how long past information can still be read out [9–11, 16]. However, to quantify not only a timescale of statistical dependence, but also its strength, one has to quantify how much of a neuron's spiking depends on its *entire past*. Here, this is done with the mutual information between the spiking of a neuron and its own past [17], also called active information storage [5–7], or predictive information [18, 19].

Estimating this mutual information directly from spike recordings, however, is notoriously difficult. The reason is that statistical dependencies may reside in precise spike times, extend far into the past and contain higher-order dependencies. This makes it hard to find a parametric model, e.g. from the family of generalized linear models [20, 21], that is flexible enough to account for the variety of spiking statistics encountered in experiments. Therefore, one

typically infers mutual information directly from observed spike trains [22–26]. The downside is that this requires a lot of data, otherwise estimates can be severely biased [27, 28]. A lot of work has been devoted to finding less biased estimates, either by correcting bias [28–31], or by using Bayesian inference [32–34]. Although these estimators alleviate to some extent the problem of bias, a reliable estimation is only possible for a much reduced representation of past spiking, also called past embedding [35]. For example, many studies infer history dependence and transfer entropy by embedding the past spiking using a single bin [26, 36].

While previously most attention was devoted to a robust estimation given a (potentially limited) embedding, we argue that a careful embedding of past activity is crucial. In particular, a past embedding should be well adapted to the spiking statistics of a neuron, but also be low-dimensional enough to enable a reliable estimation. To that end, we here devise an embedding optimization scheme that selects the embedding that maximizes the estimated history dependence, while reliable estimation is ensured by two independent regularization methods.

In this paper, we first provide a methods summary where we introduce the measure of history dependence and the information timescale, as well as the embedding optimization method employed to estimate history dependence in neural spike trains. A glossary of all the abbreviations and symbols used in this paper can be found at the beginning of Materials and methods. In Results, we first compare the measure of history dependence with classical time-lagged measures of temporal dependence on different models of neural spiking activity. Second, we test the embedding optimization approach on a tractable benchmark model, and also compare it to existing estimation methods on a variety of experimental spike recordings. Finally, we demonstrate that the approach reveals interesting differences between neural systems, both in terms of the total history dependence, as well as the information timescale. For the reader interested in applying the method, we provide practical guidelines in the discussion and in the end of Materials and methods. The method is readily applicable to highly parallel spike recordings, and a toolbox for Python3 is available online [37].

## Methods summary

### Definition of history dependence

First, we define history dependence $R(T)$ in the spiking of a single neuron. We quantify history dependence based on the mutual information

$$I(\text{spiking}; \text{past}(T)) = H(\text{spiking}) - H(\text{spiking}|\text{past}(T)) \tag{1}$$

between current spiking in a time bin $[t, t + \Delta t)$ and its own past in a past range $[t - T, t)$ (Fig 1B). Here, we assume stationarity and ergodicity, hence the measure is an average over all times $t$. This mutual information is also called active information storage [5], and is related to the predictive information [18, 19]. It quantifies how much of the current spiking information $H(\text{spiking})$ can be predicted from past spiking. The spiking information is given by the Shannon entropy [38]

$$H(\text{spiking}) = -p(\text{spike}) \log_2 p(\text{spike}) - (1 - p(\text{spike})) \log_2 (1 - p(\text{spike})), \tag{2}$$

where $p(\text{spike}) = r\Delta t$ is the probability to spike within a small time bin $\Delta t$ for a neuron with average firing rate $r$. The Shannon entropy $H(\text{spiking})$ quantifies the average information that a spiking neuron could transmit within one bin, assuming no statistical dependencies on its own past. In contrast, the conditional entropy $H(\text{spiking}|\text{past}(T))$ (see Materials and methods) quantifies the average spiking information (in the sense of entropy) that remains when dependencies on past spiking are taken into account. Note that past dependencies can only reduce the average spiking information, i.e. $H(\text{spiking}|\text{past}(T)) \leq H(\text{spiking})$. The difference between

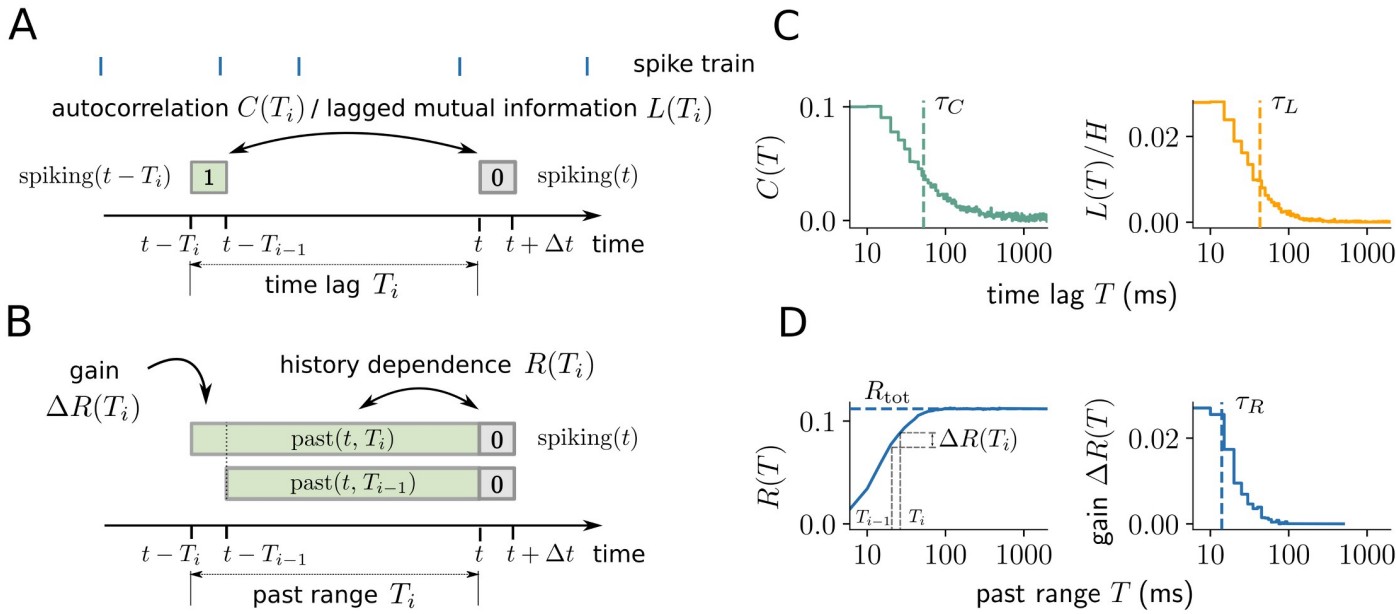

**Fig 1. Illustration of history dependence and related measures in a neural spike train.** (A) For the analysis, spiking is represented by 0 or 1 in a small time bin $\Delta t$ (grey box). Autocorrelation $C(T_i)$ or the lagged mutual information $L(T_i)$ quantify the statistical dependence of spiking on past spiking in a single past bin with time lag $T_i$ (green box). (B) In contrast, history dependence $R(T_i)$ quantifies the dependence of spiking on the entire spiking history in a past range $T_i$. The gain in history dependence $\Delta R(T_i) = R(T_i) - R(T_{i-1})$ quantifies the increase in history dependence by increasing the past range from $T_{i-1}$ to $T_i$, and is defined in analogy to the lagged measures. (C) Autocorrelation $C(T)$ and lagged mutual information $L(T)$ for a typical example neuron (mouse, primary visual cortex). Both measures decay with increasing $T$, where $L(T)$ decays slightly faster due to the non-linearity of the mutual information. Timescales $\tau_C$ and $\tau_L$ (vertical dashed lines) can be computed either by fitting an exponential decay (autocorrelation) or by using the generalized timescale (lagged mutual information). (D) In contrast, history dependence $R(T)$ increases monotonically for systematically increasing past range $T$, until it saturates at the total history dependence $R_{\text{tot}}$. From $R(T)$, the gain $\Delta R(T_i)$ can be computed between increasing past ranges $T_{i-1}$ and $T_i$ (grey dashed lines). The gain $\Delta R(T)$ decays to zero like the time-lagged measures, with information timescale $\tau_R$ (dashed line).

the two then gives the amount of spiking information that is redundant or entirely predictable from the past. To transform this measure of information into a measure of statistical dependence, we normalize the mutual information by the entropy $H(\text{spiking})$ and define history dependence $R(T)$ as

$$R(T) \equiv \frac{I(\text{spiking}; \text{past}(T))}{H(\text{spiking})} = 1 - \frac{H(\text{spiking}|\text{past}(T))}{H(\text{spiking})} \in [0, 1]. \qquad (3)$$

While the mutual information quantifies the *amount* of predictable information, $R(T)$ gives the *proportion* of spiking information that is predictable or redundant with past spiking. As such, it interpolates between the following intuitive extreme cases: $R(T) = 0$ corresponds to independent and $R(T) = 1$ to entirely predictable spiking. Moreover, while the entropy and thus the mutual information $I(\text{spiking};\text{past}(T))$ increases with the firing rate (see S13 Fig for an example on real data), the normalized $R(T)$ is comparable across recordings of neurons with very different firing rates. Finally, all the above measures can depend on the size of the time bin $\Delta t$, which discretizes the current spiking activity in time. Too small a time bin holds the risk that noise in the spike emission reduces the overall predictability or history dependence, whereas an overly large time bin holds the risk of destroying coding relevant time information in the neuron's spike train. Thus, we chose the smallest time bin $\Delta t = 5$ ms that does not yet show a decrease in history dependence (S16 Fig).

## Total history dependence and the information timescale

Here, we introduce measures to quantify the strength and the timescale of history dependence independently. First, note that the history dependence $R(T)$ monotonically increases with the past range $T$ (Fig 1D), until it converges to the *total history dependence*

$$R_{\text{tot}} \equiv \lim_{T \to \infty} R(T). \tag{4}$$

The total history dependence $R_{\text{tot}}$ quantifies the proportion of predictable spiking information once the entire past is taken into account.

While the history dependence $R(T)$ is monotonously increasing, the *gain* in history dependence $\Delta R(T_i) \equiv R(T_i) - R(T_{i-1})$ between two past ranges $T_i > T_{i-1}$ tends to decrease, and eventually decreases to zero for $T_i, T_{i-1} \to \infty$ (Fig 1D). This is in analogy to time-lagged measures of temporal dependence such as the autocorrelation $C(T)$ or lagged mutual information $L(T)$ (Fig 1A and 1C). Moreover, because $R(T)$ is monotonically increasing, the gain cannot be negative, i.e. $\Delta R(T_i) \geq 0$. From $\Delta R(T_i)$, we quantify a characteristic timescale $\tau_R$ of history dependence similar to an autocorrelation time. In analogy to the integrated autocorrelation time [39], we define the *generalized timescale*

$$\tau_R \equiv \sum_{i=1}^{n} \bar{T}_i \frac{\Delta R(T_i)}{\sum_{j=1}^{n} \Delta R(T_j)} - T_0. \tag{5}$$

as the average of past ranges $\bar{T}_i = (T_i + T_{i-1})/2$, weighted with their gain $\Delta R(T_i) = R(T_i) - R(T_{i-1})$. Here, steps between two past ranges $T_{i-1}$ and $T_i$ should be chosen small enough, and summing the middle points $\bar{T}_i$ of the steps further reduces the error of discretization. $T_0$ is the starting point, i.e. is the first past range for which $R(T)$ is computed, and was set to $T_0 = 10$ ms to exclude short-term past dependencies like refractoriness (see Materials and methods for details). Moreover, the last past range $T_n$ has to be high enough such that $R(T_n)$ has converged, i.e. $R(T_n) = R_{\text{tot}}$. Here, we set $T_n = 5$ s unless stated otherwise.

To illustrate the analogy to the autocorrelation time, we note that if the gain decays exponentially, i.e. $\Delta R(T_i) \propto \exp\left(-\frac{T_i}{\tau_{\text{auto}}}\right)$ with decay constant $\tau_{\text{auto}}$, then $\tau_R = \tau_{\text{auto}}$ for $n \to \infty$ and sufficiently small steps $T_i - T_{i-1}$. The advantage of $\tau_R$ is that it also generalizes to cases where the decay is not exponential. Furthermore, it can be applied to any other measure of temporal dependence (e.g. the lagged mutual information) as long as the sum in Eq (5) remains finite, and the coefficients are non-negative. Note that *estimates* of $\Delta R(T_i)$ can also be negative, so we included corrections to allow a sensible estimation of $\tau_R$ (Materials and methods). Finally, since $\tau_R$ quantifies the timescale over which unique predictive information is accumulated, we refer to it as the *information timescale*.

## Binary past embedding of spiking activity

In practice, estimating history dependence $R$ from spike recordings is extremely challenging. In fact, if data is limited, a reliable estimation of history dependence is only possible for a reduced representation of past spiking, also called past embedding [35]. Here, we outline how we embed past spiking activity to estimate history dependence from neural spike recordings.

First, we choose a past range $T$, which defines the time span of the past embedding. For each point in time $t$, we partition the immediate past window $[t - T, t)$ into $d$ bins and count the number of spikes in each bin. The number of bins $d$ sets the temporal resolution of the embedding. In addition, we let bin sizes scale exponentially with the bin index $j = 1, \ldots, d$ as $\tau_j = \tau_1 10^{(j-1)\kappa}$ (Fig 2A). A scaling exponent of $\kappa = 0$ translates into equal bin sizes, whereas for

**A  Uniform and exponential past embeddings for given past range $T$.**

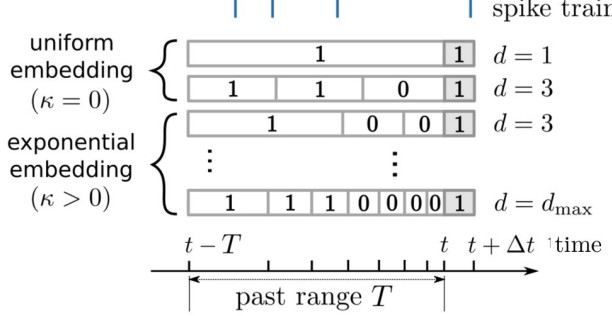

**B  Estimation of history dependence from binary spike sequences.**

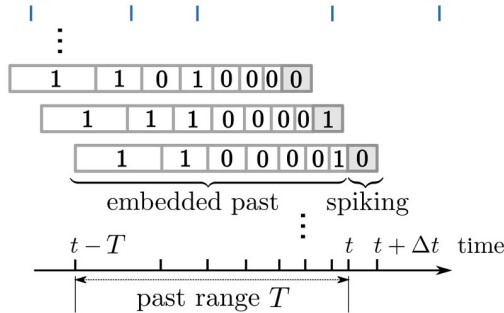

**C  Maximizing regularized estimates yields optimal past embedding for given $T$.**

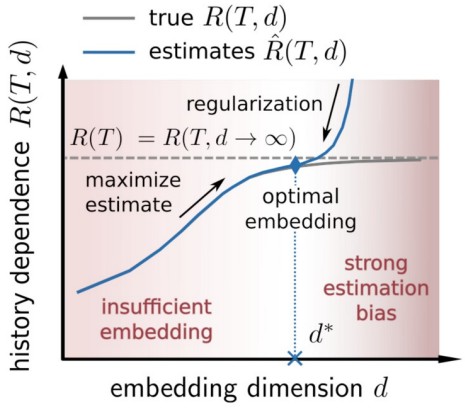

**D  Embedding-optimized estimation of history dependence and the information timescale.**

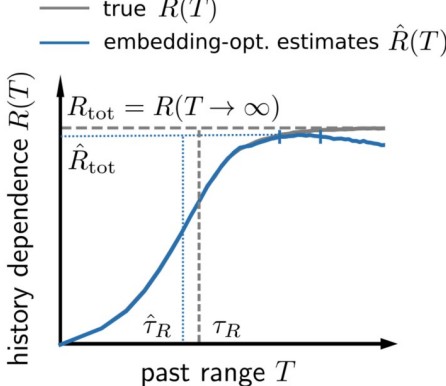

**Fig 2. Illustration of embedding optimization to estimate history dependence and the information timescale.** (A) History dependence $R$ is estimated from the observed joint statistics of current spiking in a small time bin $[t + \Delta t)$ (dark grey) and the embedded past, i.e. a binary sequence representing past spiking in a past window $[t - T, t)$. We systematically vary the number of bins $d$ and bin sizes for fixed past range $T$. Bin sizes scale exponentially with bin index and a scaling exponent $\kappa$ to reduce resolution for spikes farther into the past. (B) The joint statistics of current and past spiking are obtained by shifting the past range in steps of $\Delta t$ and counting the resulting binary sequences. (C) Finding a good choice of embedding parameters (e.g. embedding dimension $d$) is challenging: When $d$ is chosen too small, the true history dependence $R(T)$ (dashed line) is not captured appropriately (insufficient embedding) and underestimated by estimates $\hat{R}(T, d)$ (blue solid line). When $d$ is chosen too high, estimates $\hat{R}(T, d)$ are severely biased and $R(T, d)$, as well as $R(T)$, are overestimated (biased regime). Past-embedding optimization finds the optimal embedding parameter $d^*$ that maximizes the estimated history dependence $\hat{R}(T, d)$ subject to regularization. This yields a best estimate $\hat{R}(T)$ of $R(T)$ (blue diamond). (D) Estimation of history dependence $R(T)$ as a function of past range $T$. For each past range $T$, embedding parameters $d$ and $\kappa$ are optimized to yield an embedding-optimized estimate $\hat{R}(T)$. From estimates $\hat{R}(T)$, we obtain estimates $\hat{\tau}_R$ and $\hat{R}_{\text{tot}}$ of the information timescale $\tau_R$ and total history dependence $R_{\text{tot}}$ (vertical and horizontal dashed lines). To compute $\hat{R}_{\text{tot}}$ we average estimates $\hat{R}(T)$ in an interval $[T_D, T_{\text{max}}]$, for which estimates $\hat{R}(T)$ reach a plateau (vertical blue bars, see Materials and methods). For high past ranges $T$, estimates $\hat{R}(T)$ may decrease because a reliable estimation requires past embeddings with reduced temporal resolution.

$\kappa > 0$ bin sizes increase. For fixed $d$, this allows to obtain a higher temporal resolution on recent past spikes by decreasing the resolution on distant past spikes.

The past window $[t - T, t)$ of the embedding is slid forward in steps of $\Delta t$ through the whole recording with recording length $T_{\text{rec}}$, starting at $t = T$. This gives rise to $N = (T_{\text{rec}} - T)/\Delta t$ measurements of current spiking in $[t, t + \Delta t)$, and of the number of spikes in each of the $d$ past

bins ([Fig 2B]). We chose to use only binary sequences of spike counts to estimate history dependence. To that end, a count of 1 was chosen for a spike count larger than the median spike count over the $N$ measurements in the respective past bin. A binary representation drastically reduces the number of possible past sequences for given number of bins $d$, thus enabling an estimation of history dependence even from short recordings.

## Estimation of history dependence with binary past embeddings

To estimate history dependence $R$, one has to estimate the probability of a spike occurring together with different past sequences. The probabilities $\pi_i$ of these different joint events $i$ can be directly inferred from the frequencies $n_i$ with which the events occurred during the recording. Without any additional assumptions, the simplest way to estimate the probabilities is to compute the relative frequencies $\hat{\pi}_i = n_i/N$, where $N$ is the total number of observed joint events. This estimate is the maximum likelihood (ML) estimate of joint probabilities $\pi_i$ for a multinomial likelihood, and the corresponding estimate of history dependence will also be denoted by ML. This direct estimate of history dependence is known to be strongly biased when data is too limited [28, 30]. The bias is typically positive, because, under limited data, probabilities of *observed* joint events are given too much weight. Therefore, statistical dependencies are overestimated. Even worse, the overestimation becomes more severe the higher the number of possible past sequences $K$. Since $K$ increases exponentially with the dimension of the past embedding $d$, i.e. $K = 2^d$ for binary spike sequences, history dependence is severely overestimated for high $d$ ([Fig 2C]). The potential overestimation makes it hard to choose embeddings that represent past spiking sufficiently well. In the following, we outline how one can optimally choose embeddings if appropriate regularization is applied.

## Estimating history dependence with past-embedding optimization

Due to systematic overestimation, high-dimensional past embeddings are prohibitive for a reliable estimation of history dependence from limited data. Yet, high-dimensional past embeddings might be required to capture all history dependence. The reason is that history dependence may reside in precise spike times, but also may extend far into the past.

To illustrate this trade-off, we consider a discrete past embedding of spiking activity in a past range $T$, where the past spikes are assigned to $d$ equally large bins ($\kappa = 0$). We would like to obtain an estimate $\hat{R}(T)$ of the maximum possible history dependence $R(T)$ for the given past range $T$, with $R(T) \equiv R(T, d \rightarrow \infty)$ ([Fig 2C]). The number of bins $d$ can go to infinity only in theory, though. In practice, we have estimates $\hat{R}(T, d)$ of the history dependence $R(T, d)$ for finite $d$. On the one hand, one would like to choose a high number of bins $d$, such that $R(T, d)$ approximates $R(T)$ well for the given past range $T$. Too few bins $d$ otherwise reduce the temporal resolution, such that $R(T, d)$ is substantially less than $R(T)$ ([Fig 2C]). On the other hand, one would like to choose $d$ not too large in order to enable a reliable estimation from limited data. If $d$ is too high, estimates $\hat{R}(T, d)$ strongly overestimate the true history dependence $R(T, d)$ ([Fig 2C]).

Therefore, if the past embedding is not chosen carefully, history dependence is either overestimated due to strong estimation bias, or underestimated because the chosen past embedding was too simple.

Here, we thus propose the following *past-embedding optimization* approach: For a given past range $T$, select embedding parameters $d^*$, $\kappa^*$ that maximize the estimated history dependence $\hat{R}(T, d, \kappa)$, while overestimation is avoided by an appropriate regularization. This yields an embedding-optimized estimate $\hat{R}(T) = \hat{R}(T, d^*, \kappa^*)$ of the true history dependence $R(T)$.

In terms of the above example, past-embedding optimization selects the optimal embedding dimension $d^*$, which provides the best lower bound $\hat{R}(T) = \hat{R}(T, d^*)$ to $R(T)$ (Fig 2C).

Since we can anyways provide only a lower bound, regularization only has to ensure that estimates $\hat{R}(T, d, \kappa)$ are either unbiased, or a lower bound to the observable history dependence $R(T, d, \kappa)$. For that purpose, in this paper we introduce a Bayesian bias criterion (BBC) that selects only unbiased estimates. In addition, we use an established bias correction, the so-called Shuffling estimator [31] that, within leading order of the sample size, is guaranteed to provide a lower bound to the observable history dependence (see Materials and methods for details).

Together with these regularization methods, the embedding optimization approach enables complex embeddings of past activity while minimizing the risk of overestimation. See Materials and methods for details on how we used embedding-optimized estimates $\hat{R}(T)$ to compute estimates $\hat{R}_{\text{tot}}$ and $\hat{\tau}_R$ of the total history dependence and information timescale (Fig 2D, blue dashed lines).

## Results

In the first part, we demonstrate the differences between history dependence and time-lagged measures of temporal dependence for several models of neural spiking activity. We then benchmark the estimation of history dependence using embedding optimization on a tractable neuron model with long-lasting spike adaptation. Moreover, we compare the embedding optimization approach to existing estimation methods on a variety of extra-cellular spike recordings. In the last part, we apply this to analyze history dependence for a variety of neural systems, and compare the results to the autocorrelation and other statistical measures on the data.

### Differences between history dependence and time-lagged measures of temporal dependence

The history dependence $R(T)$ quantifies how predictable neural spiking is, given activity in a certain past range $T$. In contrast, time-lagged measures of temporal dependence like the autocorrelation $C(T)$ [40] or lagged mutual information $L(T)$ [41, 42] quantify the dependence of spiking on activity in a single past bin with time lag $T$ (Fig 1A and 1C; Materials and methods). In the following, we showcase the main differences between the two approaches.

**History dependence disentangles the effects of input activation, reactivation and temporal depth of a binary autoregressive process.** To show the behavior of the measures in a well controlled setup, we analyzed a simple binary autoregressive process with varying temporal depth $l$ (Fig 3A). The process evolves in discrete time steps, and has an active (1) or inactive (0) state. Active states are evoked either by external input with probability $h$, or by internal reactivations that are triggered by activity within the past $l$ steps. Each past activation increases the reactivation probability by $m$, which regulates the strength of history dependence in the process. In the following, we describe how the measures behave as we vary each of the different model parameters, and then summarize the key difference between the measures.

The input strength $h$ increases the firing rate and thus the spiking entropy $H(\text{spiking})$. This leads to a strong increase in the total mutual information $I_{\text{tot}} \equiv \lim_{T \to \infty} I(\text{spiking}; \text{past}(T))$, whereas the total history dependence $R_{\text{tot}}$ is normalized by the entropy and does slightly decrease (Fig 3B). This slight decrease is expected from a sensible measure of history dependence, because the input is random and has no temporal dependence. In addition, input

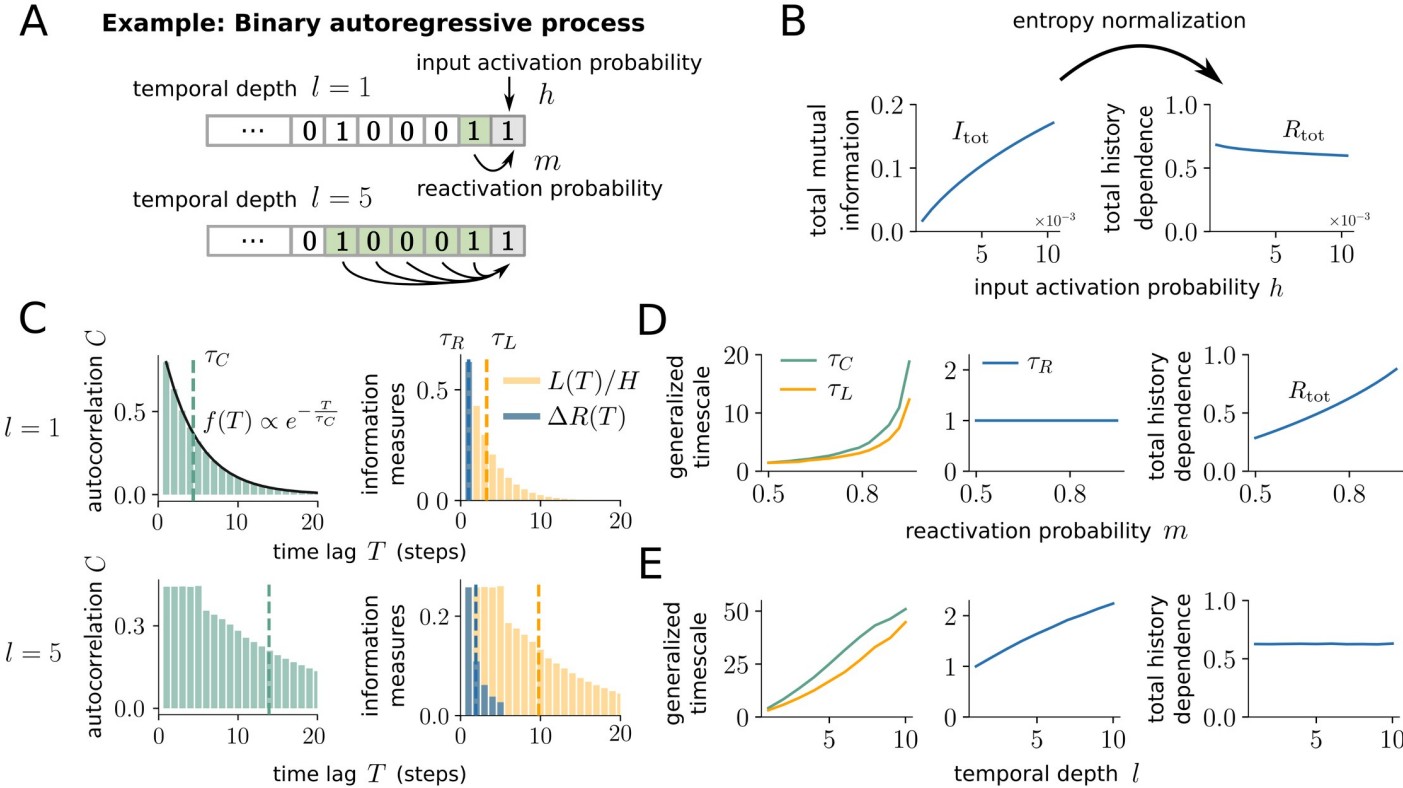

**Fig 3. History dependence disentangles the effects of input activation, reactivation and temporal depth of a binary autoregressive process.** (A) In the binary autoregressive process, the state of the next time step (grey box) is active (one) either because of an input activation with probability $h$, or because of an internal reactivation. The internal activation is triggered by activity in the past $l$ time steps (green), where each active state increases the activation probability by $m$. (B) Increasing the input activation probability $h$ increases the total mutual information $I_{\text{tot}}$, although input activations are random and therefore not predictable. Normalizing the total mutual information by the entropy yields the total history dependence $R_{\text{tot}}$, which decreases mildly with $h$. (C) Autocorrelation $C(T)$, lagged mutual information $L(T)$ and gain in history dependence $\Delta R(T)$ decay differently with the time lag $T$. For $l = 1$ and $m = 0.8$ (top), autocorrelation $C(T)$ decays exponentially with autocorrelation time $\tau_C$, whereas $L(T)$ decays faster due to the non-linearity of the mutual information. For $l = 5$ (bottom), $C(T)$ and $L(T)$ plateau over the temporal depth, and then decay much slower than for $l = 1$. In contrast, $\Delta R(T)$ is non-zero only for $T$ shorter or equal to the temporal depth of the process, with much shorter timescale $\tau_R$. Parameters $m$ and $h$ were adapted to match the firing rate and total history dependence between $l = 1$ and $l = 5$. (D) When increasing the reactivation probability $m$ for $l = 1$, timescales of time-lagged measures $\tau_C$ and $\tau_L$ increase. For history dependence, the information timescale $\tau_R$ remains constant, but the total history dependence $R_{\text{tot}}$ increases. (E) When varying the temporal depth $l$, all timescales increased. Parameters $h$ and $m$ were adapted to hold the firing rate and $R_{\text{tot}}$ constant.

activations may fall together with internal activations, which slightly reduces the total history dependence.

In contrast, the total history dependence $R_{\text{tot}}$ increases with the reactivation probability $m$, as expected (Fig 3D). For the autocorrelation, the reactivation probability $m$ not only influences the magnitude of the correlation coefficients, but also the decay of the coefficients. For autoregressive processes (and $l = 1$), autocorrelation coefficients $C(T)$ decay exponentially [14] (Fig 3C), where the autocorrelation time $\tau_C = -\Delta t / \log(m)$ increases with $m$ and diverges as $m \to 1$ (Fig 3D). The lagged mutual information $L(T)$ is a non-linear measure of time-lagged dependence, and has a very similar behavior as the autocorrelation, with a slightly faster decay and thus smaller generalized timescale $\tau_L$ (Fig 3C and 3D). Note that we normalized $L(T)$ by the spiking entropy $H$ to make it directly comparable to $\Delta R(T)$. In contrast to the time-lagged measures, the gain in history dependence $\Delta R(T)$ is only non-zero for $T$ smaller or equal to the true temporal depth $l$ of the process (Fig 3C). As a consequence, the information timescale $\tau_R$ does not increase with $m$ for fixed $l$ (Fig 3D).

Finally, the temporal depth $l$ controls how far into the past activations depend on their preceding activity. Indeed, we find that the information timescale $\tau_R$ increases with $l$ as expected (Fig 3C and 3E). Similarly, the timescales of the time-lagged measures $\tau_C$ and $\tau_L$ increase with the temporal depth $l$. Note that parameters $m$ and $h$ were adapted for each $l$ to keep the firing rate and total history dependence $R_{\text{tot}}$ constant, hence differences in the timescale can be unambiguously attributed to the increase in $l$.

To conclude, history dependence disentangles the effects of input activation, reactivation and temporal depth, which provides a comprehensive characterization of past dependencies in the autoregressive model. This is different from the total mutual information, which lacks the entropy normalization and is sensitive to the firing rate. This is also different from time-lagged measures, whose timescales are sensitive to both, the reactivation probability $m$ *and* the temporal depth $l$. The confusion of effects in the timescales is rooted in the time-lagged nature of the measures—by quantifying past dependencies out of context, $C(T)$ and $L(T)$ also capture *indirect, redundant* dependencies onto past events. Indirect, redundant dependencies arise from unique dependencies, because past states that are uniquely predictive of future activities were in turn uniquely dependent on their own past. The stronger the unique dependence, the longer the indirect dependencies reach into the past, which increases the timescale of time-lagged measures. In contrast, indirect dependencies do not contribute to the history dependence, because they add no predictive information once more-recent past is taken into account.

**History dependence dismisses redundant past dependencies and captures synergistic effects.** A key property of history dependence is that it evaluates past dependencies in the light of more-recent past. This allows the measure to dismiss indirect, redundant past dependencies and to capture synergistic effects. In three common models of neural spiking activity, we demonstrate how this leads to a substantially different characterization of past dependencies compared to time-lagged measures of temporal dependence.

First, we simulated a subsampled branching process [14], which is a minimal model for activity propagation in neural networks and captures key properties of spiking dynamics in cortex [15]. Similar to the binary autoregressive process, active neurons activate neurons in the next time step with probability $m$, the so-called branching parameter, and are activated externally with some probability $h$. The process was simulated in time steps of $\Delta t = 4$ ms with a population activity of 500 Hz, which was subsampled to obtain a single spike train with a firing rate of 5 Hz (Fig 4A). Similar to the binary autoregressive process, the autocorrelation decays exponentially with autocorrelation time $\tau_C = -\Delta t / \log(m) = 198$ ms, and the lagged mutual information decays slightly faster (Fig 4B). In comparison, the gain in history dependence $\Delta R$ decays much faster. When increasing the branching parameter $m$ (for fixed firing rate), the total history dependence increased, as in the autoregressive process (S11 Fig). Strikingly, the timescale $\tau_R$ remained constant or even decreased for larger $m > 0.967$ and thus higher autocorrelation time $\tau_C > 120$ ms (S11 Fig), which is different from the binary autoregressive process. The reason is that the branching process evolves at the population level, whereas history dependence is quantified at the single neuron level. Thereby, history dependence also captures indirect dependencies, because the own spiking history reflects the population activity. The higher the branching parameter $m$, the more informative past spikes are about the population activity, and the shorter is the timescale $\tau_R$ over which all the relevant information about the population activity can be collected. Thus, for the branching process, the total history dependence $R_{\text{tot}}$ captures the influence of the branching parameter, whereas the information timescale $\tau_R$ behaves very differently from the timescales of time-lagged measures.

Second, we demonstrate the difference of history dependence to time-lagged measures on an Izhikevich neuron, which is a flexible model that can produce different neural firing patterns similar to those observed for real neurons [44]. Here, parameters were chosen according

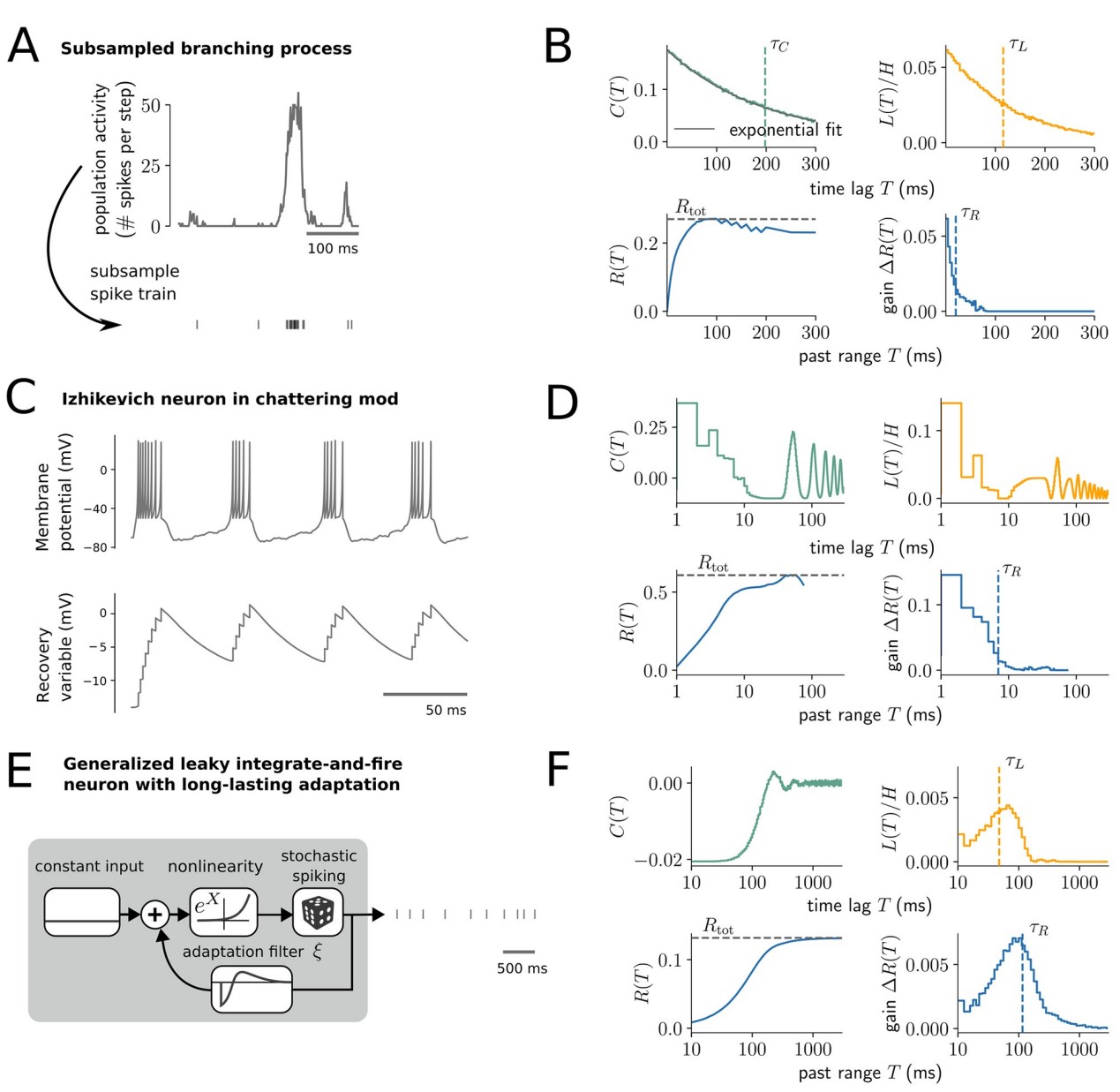

**Fig 4. History dependence dismisses redundant past dependencies and captures synergistic effects.** (A,B) Analysis of a subsampled branching process. (A) The population activity was simulated as a branching process ($m = 0.98$) and subsampled to yield the spike train of a single neuron (Materials and methods). (B) Autocorrelation $C(T)$ and lagged mutual information $L(T)$ include redundant dependencies and decay much slower than the gain $\Delta R(T)$, with much longer timescales (vertical dashed lines). (C,D) Analysis of an Izhikevich neuron in chattering mode with constant input and small voltage fluctuations. The neuron fires in regular bursts of activity. (D) Time-lagged measures $C(T)$ and $L(T)$ measure both, intra- ($T < 10$ ms) and inter-burst ($T > 10$ ms) dependencies, which decay very slowly due to regularity of the firing. The gain $\Delta R(T)$ reflects that most spiking can already be predicted from intra-burst dependencies, whereas inter-burst dependencies are highly redundant. In this case, only $\Delta R(T)$ yields a sensible time scale (blue dashed line). (E, F) Analysis of a generalized leaky integrate-and-fire neuron with long-lasting adaptation filter $\xi$ [3, 43] and constant input. (F) Here, $\Delta R(T)$ decays slower to zero than the autocorrelation $C(T)$, and is higher than $L(T)$ for long $T$. Therefore, the dependence on past spikes is stronger when taking more-recent past spikes into account ($\Delta R(T)$), as when considering them independently ($L(T)$). Due to these synergistic past dependencies, $\Delta R(T)$ is the only measure that captures the long-range nature of the spike adaptation.

 

to the "chattering mode" [44], with constant input and small voltage fluctuations (Materials and methods). The neuron fires in regular bursts of activity, with consistent timing between spikes within and between bursts (Fig 4C). While time-lagged measures capture all the regularities in spiking and oscillate with the bursts of activity, history dependence correctly captures that spiking can almost be entirely predicted from intra-burst dependencies alone (Fig 4D). History dependence dismisses the redundant inter-burst dependencies and thereby yields a sensible measure of a timescale (blue dashed line).

Finally, we analyzed a generalized leaky integrate-and-fire neuron with long-range spike adaptation (22 seconds) (Fig 4E), which reproduces spike-frequency adaptation as observed for somatosensory pyramidal neurons [3, 43]. For this model, time-lagged measures $C(T)$ and $L(T)$ actually decay to zero much faster than the gain in history dependence $\Delta R(T)$, which is the only measure that captures the long-range adaptation effects of the model (Fig 4F). This shows that past dependencies in this model include synergistic effects, where the dependence is stronger in the context of more-recent spikes. This is most likely due to the non-linearity of the model, where past spikes cause a different adaptation when taken together as when considered as the sum of their contributions.

Thus, due to its ability to dismiss redundant past dependencies and to capture synergistic effects, history dependence really provides a complementary characterization of past dependencies compared to time-lagged measures. Importantly, because the approach better disentangles the effects of timescale and total history dependence, the results remain interpretable for very different models, and provide a more comprehensive view on past dependencies.

## Embedding optimization captures history dependence for a neuron model with long-lasting spike adaptation

On a benchmark spiking neuron model, we first demonstrate that without optimization and proper regularization, past embeddings are likely to capture much less history dependence, or lead to estimates that severely overestimate the true history dependence. Readers that are familiar with the bias problem of mutual information estimation might want to jump to the next part, where we validate that embedding-optimized estimates indeed capture the model's true history dependence, while being robust to systematic overestimation. As a model we chose a generalized leaky integrate-and-fire (GLIF) model with spike frequency adaptation, whose parameters were fitted to experimental data [3, 43]. The model was chosen, because it is equipped with a long-lasting spike adaptation mechanism, and its total history dependence $R_{\text{tot}}$ can be directly computed from sufficiently long simulations (Materials and methods). For demonstration, we show results on a variant of the model where adaptation reaches one second into the past, and show results on the original model with a 22 second kernel in S1, S2 and S5 Figs. For simulation, the neuron was driven with a constant input current to achieve an average firing rate of 4 Hz. In the following, estimates $\hat{R}(T)$ are shown for a simulated recording of 90 minutes, whereas the true values $R(T)$ were computed on a 900 minute recording (Materials and methods).

**Without regularization, history dependence is severely overestimated for high-dimensional embeddings.** For demonstration, we estimated the history dependence $R(\tau, d)$ for varying numbers of bins $d$ and a constant bin size $\tau = 20$ ms (i.e. $\kappa = 0$ and $T = d \cdot \tau$). We compared estimates $\hat{R}(\tau, d)$ obtained by maximum likelihood (ML) estimation [28], or Bayesian estimation using the NSB estimator [33], with the model's true $R(\tau, d)$ (Fig 5A). Both estimators accurately estimate $R(\tau, d)$ for up to $d \approx 20$ past bins. As expected, the NSB estimator starts to be biased at higher $d$ than the ML estimator. For embedding dimensions $d > 30$, both estimators severely overestimate $R(\tau, d)$. Note that $\pm$ two standard deviations are plotted as shaded

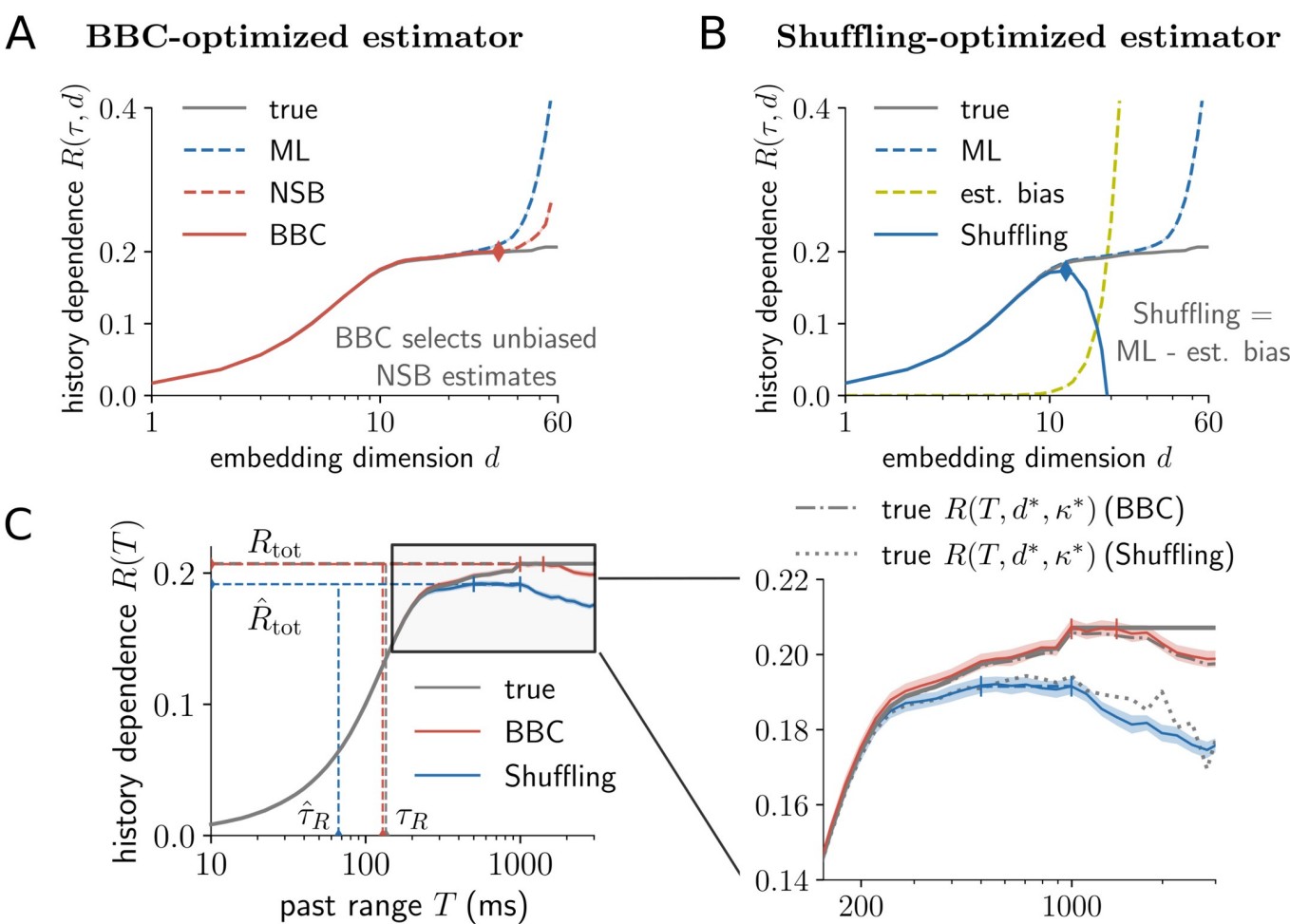

**Fig 5. Embedding optimization captures history dependence for a neuron model with long-lasting spike adaptation.** Results are shown for a generalized leaky integrate-and-fire (GLIF) model with long-lasting spike frequency adaptation [3, 43] with a temporal depth of one second (Methods and material). (A) For illustration, history dependence $R(\tau, d)$ was estimated on a simulated 90 minute recording for different embedding dimensions $d$ and a fixed bin width $\tau = 20$ ms. Maximum likelihood (ML) [28] and Bayesian (NSB) [33] estimators display the insufficient embedding versus estimation bias trade-off: For small embedding dimensions $d$, the estimated history dependence is much smaller, but agrees well with the true history dependence $R(\tau, d)$ for the given embedding. For larger $d$, the estimated history dependence $\hat{R}(\tau, d)$ increases, but when $d$ is too high ($d > 20$), it severely overestimates the true $R(\tau, d)$. The Bayesian bias criterion (BBC) selects NSB estimates $\hat{R}(\tau, d)$ for which the difference between ML and NSB estimate is small (red solid line). All selected estimates are unbiased and agree well with the true $R(\tau, d)$ (grey line). Embedding optimization selects the highest, yet unbiased estimate (red diamond). (B) The Shuffling estimator (blue solid line) subtracts estimation bias on surrogate data (yellow dashed line) from the ML estimator (blue dashed line). Since the surrogate bias is higher than the systematic overestimation in the ML estimator (difference between grey and blue dashed lines), the Shuffling estimator is a lower bound to $R(\tau, d)$. Embedding optimization selects the highest estimate, which is still a lower bound (blue diamond). For A and B, shaded areas indicate ± two standard deviations obtained from 50 repeated simulations, which are very small and thus hardly visible. (C) Embedding-optimized BBC estimates $\hat{R}(T)$ (red line) yield accurate estimates of the model neuron's true history dependence $R(T)$, total history dependence $R_{\text{tot}}$ and information timescale $\tau_R$ (horizontal and vertical dashed lines). The zoom-in (right panel) shows robustness of both regularization methods: For all $T$ the model neuron's $R(T, d^*, \kappa^*)$ lies within errorbars (BBC), or consistently above the Shuffling estimator that provides a lower bound. Here, the model's $R(T, d^*, \kappa^*)$ was computed for the optimized embedding parameters $d^*, \kappa^*$ that were selected via BBC or Shuffling, respectively (dashed lines). Shaded areas indicate ± two standard deviations obtained by bootstrapping, and colored vertical bars indicate past ranges over which estimates $\hat{R}(T)$ were averaged to compute $\hat{R}_{\text{tot}}$ (Materials and methods).

areas, but are too small to be visible. Therefore, any deviation of estimates from the model's true history dependence $R(\tau, d)$ can be attributed to positive estimation bias, i.e. a systematic overestimation of the true history dependence due to limited data.

The aim is now to identify the largest embedding dimension $d^*$ for which the estimate of $R(\tau, d)$ is not yet biased. A biased estimate is expected as soon as the two estimates ML and

NSB start to differ significantly from each other (Fig 5A, red diamond), which is formalized by the Bayesian bias criterion (BBC) (Materials and methods). According to the BBC, all NSB estimates $\hat{R}(\tau, d)$ with $d$ lower or equal to $d^*$ are unbiased (solid red line). We find that indeed all BBC estimates agree well with the true $R(\tau, d)$ (grey line), but $d^*$ yields the largest unbiased estimate.

The problem of estimation bias has also been addressed previously by the so-called Shuffling estimator [31]. The Shuffling estimator is based on the ML estimator and applies a bias correction term (Fig 5B). In detail, one approximates the estimation bias using surrogate data, which are obtained by shuffling of the embedded spiking history. The surrogate estimation bias (yellow dashed line) is proven to be larger than the actual estimation bias (difference between grey solid and blue dashed line). Therefore, Shuffling estimates $\hat{R}(\tau, d)$ provide lower bounds to the true history dependence $R(\tau, d)$. As with the BBC, one can safely maximize Shuffling estimates $\hat{R}(\tau, d)$ over $d$ to find the embedding dimension $d^*$ that provides the largest lower bound to the model's total history dependence $R_{tot}$ (Fig 5B, blue diamond).

Thus, using a model neuron, we illustrated that history dependence can be severely overestimated if the embedding is chosen too complex. Only when overestimation is tamed by one of the two regularization methods, BBC or Shuffling, embedding parameters can be safely optimized to yield better estimates of history dependence.

**Optimized embeddings capture the model's true history dependence.** In the previous part, we demonstrated how embedding parameters are optimized for the example of fixed $\kappa$ and $\tau$. Now, we optimize all embedding parameters for fixed past range $T$ to obtain embedding-optimized estimates $\hat{R}(T)$ of $R(T)$. We find that embedding-optimized BBC estimates $\hat{R}(T)$ agree well with the true $R(T)$, hence the model's total history dependence $R_{tot}$ and information timescale $\tau_R$ are well estimated (Fig 5C, vertical and horizontal dashed lines). In contrast, the Shuffling estimator underestimates the true $R(T)$ for past ranges $T > 200$ ms, hence the model's $R_{tot}$ and $\tau_R$ are underestimated (blue dashed lines). For large past ranges $T > 1000$ ms, estimates $\hat{R}(T)$ of both estimators decrease again, because no additional history dependence is uncovered, whereas the constraint of an unbiased estimation decreases the temporal resolution of the embedding.

**Embedding-optimized estimates are robust to overestimation despite maximization over complex embeddings.** In the previous part, we investigated how much of the true history dependence for different past ranges $T$ (grey solid line) we miss by embedding the spiking history. An additional source of error is the estimation of history dependence from limited data. In particular, estimates are prone to overestimate history dependence systematically (Fig 5A and 5B).

To test explicitly for overestimation, we computed the true history dependence $R(T, d^*, \kappa^*)$ for exactly the same sets of embedding parameters $T, d^*, \kappa^*$ that were found during embedding optimization with BBC (grey dash-dotted line), and the Shuffling estimator (grey dotted line, Fig 5C, zoom-in). We expect that BBC estimates are unbiased, i.e. the true history dependence should lie within errorbars of the BBC estimates (red shaded area) for a given $T$. In contrast, Shuffling estimates are a lower bound, i.e. estimates should lie below the true history dependence (given the same $T, d^*, \kappa^*$). We find that this is indeed the case for all $T$. Note that this is a strong result, because it requires that the regularization methods work reliably for every single set of embedding parameters used for optimization—otherwise, parameters that cause overestimation would be selected.

Thus, we can confirm that the embedding-optimized estimates do not systematically overestimate the model neuron's history dependence, and are on average lower bounds to the true history dependence. This is important for the interpretation of the results.

**Mild overfitting can occur during embedding optimization on short recordings, but can be overcome with cross-validation.** We also tested whether the recording length affects the reliability of embedding-optimized estimates, and found very mild overestimation (1–3%) of history dependence for BBC for recordings as short as 3 minutes (S1 and S4 Figs). The overestimation is a consequence of overfitting during embedding optimization: Variance in the estimates increases for shorter recordings, hence maximizing over estimates selects embedding parameters that have high history dependence by chance. Therefore, the overestimation can be overcome by cross-validation, e.g. by optimizing embedding parameters on the first half, and computing estimates on the second half of the data (S1 Fig). In contrast, we found that for the model neuron, Shuffling estimates do not overestimate the true history dependence even for recordings as short as 3 minutes (S1 Fig). This is because the effect of overfitting was small compared to the systematic underestimation of Shuffling estimates. Here, all experimental recordings where we apply BBC are long enough ($\approx$ 90 minutes), thus no cross-validation was applied on the experimental data.

**Estimates of the information timescale are sensitive to the recording length.** Finally, we also tested the impact of the recording length on estimates $\hat{R}_{\text{tot}}$ of the total history dependence as well as estimates $\hat{\tau}_R$ of the information timescale. While on recordings of 3 minutes embedding optimization still estimated $\approx$ 95% of the true $R_{\text{tot}}$, estimates $\hat{\tau}_R$ were only $\approx$ 75% of the true $\tau_R$ (S2 Fig). Thus, estimates of the information timescale $\tau_R$ are more sensitive to the recording length, because they depend on the small additional contributions to $R(T)$ for high past ranges $T$, which are hard to estimate for short recordings. Therefore, we advice to analyze recordings of similar length to make results on $\tau_R$ comparable across experiments. In the following, we explicitly shorten some recordings such that all recordings have approximately the same recording length.

In conclusion, embedding optimization accurately estimated the model neuron's true history dependence. Moreover, for all past ranges, embedding-optimized estimates were robust to systematic overestimation. Embedding optimization is thus a promising approach to quantify history dependence and the information timescale in experimental spike recordings.

## Embedding optimization is key to estimate long-lasting history dependence in extra-cellular spike recordings

Here, we apply embedding optimization to long spike recordings ($\approx$ 90 minutes) from rat dorsal hippocampus layer CA1 [45, 46], salamander retina [47, 48] and in vitro recordings of rat cortical culture [49]. In particular, we compare embedding optimization to other popular estimation approaches, and demonstrate that an exponential past embedding is necessary to estimate history dependence for long past ranges.

**Embedding optimization reveals history dependence that is not captured by a generalized linear model or a single past bin.** We use embedding optimization to estimate history dependence $R(T)$ as a function of the past range $T$ (see Fig 6B for an example single unit from hippocampus layer CA1, and S6, S7 and S8 Figs for all analyzed sorted units). In this example, BBC and Shuffling with a maximum of $d_{\max} = 20$ past bins led to very similar estimates for all $T$. Notably, embedding optimization with both regularization methods estimated high total history dependence of almost $R_{\text{tot}} \approx 40\%$ with a temporal depth of almost a second, and an information timescale of $\tau_R \approx 100$ ms (Fig 6B). This indicates that embedding-optimized estimates capture a substantial part of history dependence also in experimental spike recordings.

Importantly, other common estimation approaches fail to capture the same amount of history dependence (Fig 6B and 6D). To compare how well the different estimation approaches could capture the total history dependence, we plotted for each sorted unit the different

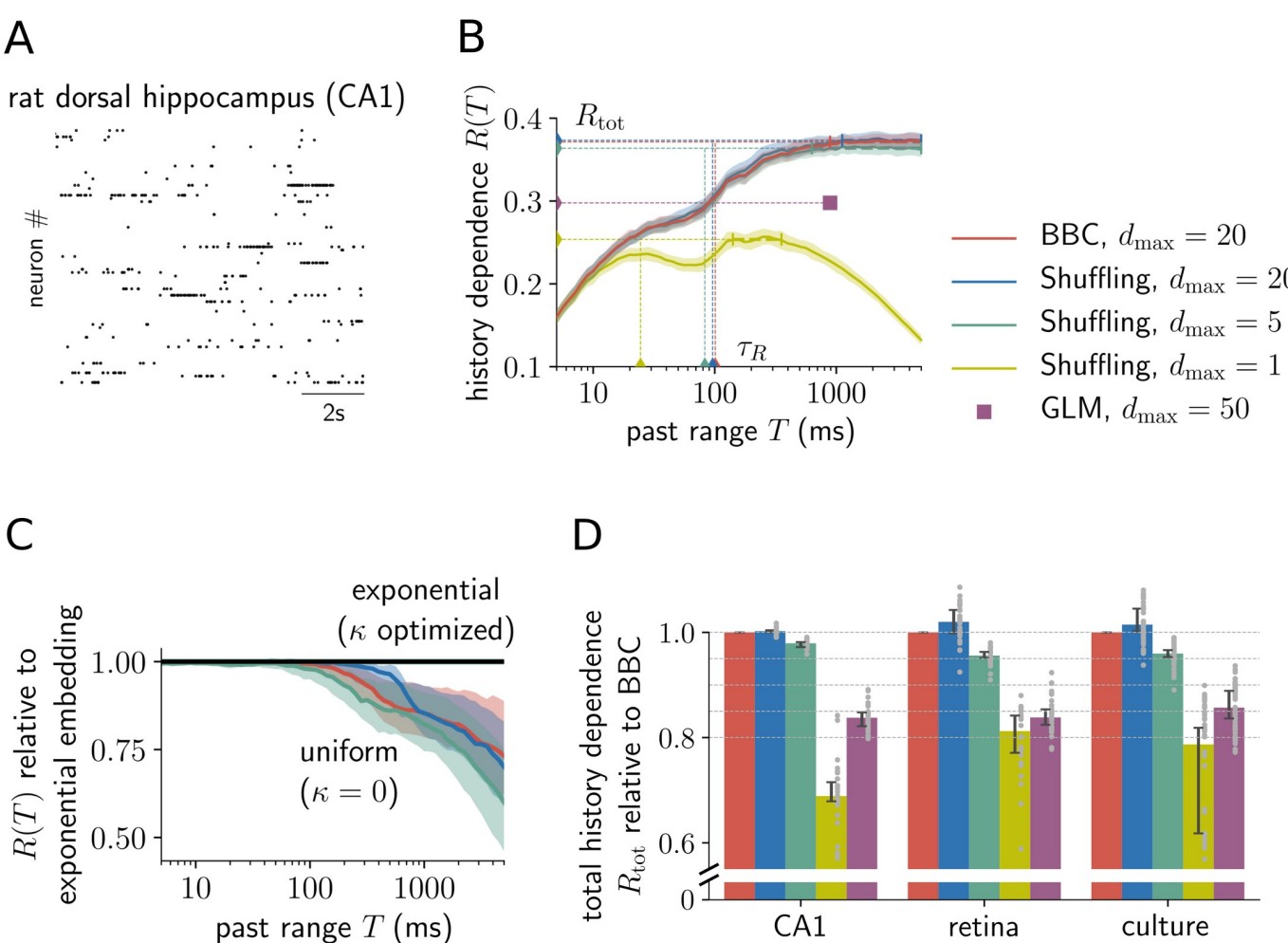

**Fig 6. Embedding optimization is key to estimate long-lasting history dependence in extra-cellular spike recordings.** (A) Example of recorded spiking activity in rat dorsal hippocampus layer CA1. (B) Estimates of history dependence $R(T)$ for various estimators, as well as estimates of the total history dependence $R_{tot}$ and information timescale $\tau_R$ (dashed lines) (example single unit from CA1). Embedding optimization with BBC (red) and Shuffling (blue) for $d_{max} = 20$ yields consistent estimates. Embedding-optimized Shuffling estimates with a maximum of $d_{max} = 5$ past bins (green) are very similar to estimates obtained with $d_{max} = 20$ (blue). In contrast, using a single past bin ($d_{max} = 1$, yellow), or fitting a GLM for the temporal depth found with BBC (violet dot), estimates much lower total history dependence. Shaded areas indicate ± two standard deviations obtained by bootstrapping, and small vertical bars indicate past ranges over which estimates of $R(T)$ were averaged to estimate $R_{tot}$ (Materials and methods). (C) An exponential past embedding is crucial to capture history dependence for high past ranges $T$. For $T > 100$ ms, uniform embeddings strongly underestimate history dependence. Shown is the median of embedding-optimized estimates of $R(T)$ with uniform embeddings, relative to estimates obtained by optimizing exponential embeddings, for BBC with $d_{max} = 20$ (red) and Shuffling with $d_{max} = 20$ (blue) and $d_{max} = 5$ (green). Shaded areas show 95% percentiles. Median and percentiles were computed over analyzed sorted units in CA1 ($n = 28$). (D) Comparison of total history dependence $R_{tot}$ for different estimation and embedding techniques for three different experimental recordings. For each sorted unit (grey dots), estimates are plotted relative to embedding-optimized estimates for BBC and $d_{max} = 20$. Embedding optimization with Shuffling and $d_{max} = 20$ yields consistent but slightly higher estimates than BBC. Strikingly, Shuffling estimates for as little as $d_{max} = 5$ past bins (green) capture more than 95% of the estimates for $d_{max} = 20$ (BBC). In contrast, estimates obtained by optimizing a single past bin, or fitting a GLM, are considerably lower. Bars indicate the median and lines indicate 95% bootstrap confidence intervals on the median over analyzed sorted units (CA1: $n = 28$; retina: $n = 111$; culture: $n = 48$).

estimates of $R_{tot}$ relative to the corresponding BBC estimate (Fig 6D). Embedding optimization with Shuffling yields estimates that agree well with BBC estimates. The Shuffling estimator even yields slightly higher values on the experimental data. Interestingly, embedding optimization with the Shuffling estimator and as little as $d_{max} = 5$ past bins captures almost the same history dependence as BBC with $d_{max} = 20$, with a median above 95% for all neural systems. In contrast, we find that a single past bin only accounts for 70% to 80% of the total history

dependence. A GLM bears little additional advantage with a slightly higher median of $\approx 85\%$. To save computation time, GLM estimates were only computed for the temporal depth that was estimated using BBC (Fig 6B, violet square). The remaining embedding parameters $d$ and $\kappa$ of the GLM's history kernel were separately optimized using the Bayesian information criterion (Materials and methods). Since parameters were optimized, we argue that the GLM underestimates history dependence because of its specific model assumptions, i.e. no interactions between past spikes. Moreover, we found that the GLM performs worse than embedding optimization with only five past bins. Therefore, we conclude that for typical experimental spike trains, interactions between past spikes are important, but do not require very high temporal resolution. In the remainder of this paper we use the reduced approach (Shuffling $d_{\max} = 5$) to compare history dependence among different neural systems.

**Increasing bin sizes exponentially is crucial to estimate long-lasting history dependence.** To demonstrate this, we plotted embedding-optimized BBC estimates of $R(T)$ using a uniform embedding, i.e. equal bin sizes, relative to estimates obtained with exponential embedding (Fig 6C), both for BBC with $d_{\max} = 20$ (red) and Shuffling with $d_{\max} = 20$ (blue) or $d_{\max} = 5$ (green). For past ranges $T > 100$ ms, estimates using a uniform embedding miss considerable history dependence, which becomes more severe the longer the past range. In the case of $d_{\max} = 5$, a uniform embedding captures around 80% for $T = 1$ s, and only around 60% for $T = 5$ s (median over analyzed sorted units in CA1). Therefore, we argue that an exponential embedding is crucial for estimating long-lasting history dependence.

## Together, total history dependence and the information timescale show clear differences between neural systems

In addition to recordings from rat dorsal hippocampus layer CA1, salamander retina and rat cortical culture, we analyzed sorted units in a recording of mouse primary visual cortex using the novel Neuropixels probe [50]. Recordings from primary visual cortex were approximately 40 minutes long. Thus, to make results comparable, we analyzed only the first 40 minutes of all recordings.

We find clear differences between the neural systems, both in terms of the total history dependence, as well as the information timescale (Fig 7A). Sorted units in cortical culture and hippocampus layer CA1 have high total history dependence $R_{\text{tot}}$ with median over sorted units of $\approx 24\%$ and $\approx 25\%$, whereas sorted units in retina and primary visual cortex have typically low $R_{\text{tot}}$ of $\approx 11\%$ and $\approx 8\%$. In terms of the information timescale $\tau_R$, sorted units in hippocampus layer CA1 display much higher $\tau_R$, with a median of $\approx 96$ ms, than units in cortical culture, with median $\tau_R$ of $\approx 12$ ms. Similarly, sorted units in primary visual cortex have higher $\tau_R$, with median of $\approx 37$ ms, than units in retina, with median of $\approx 23$ ms. These differences could reflect differences between early visual processing (retina, primary visual cortex) and high level processing and memory formation in hippocampus, or likewise, between neural networks that are mainly input driven (retina) or exclusively driven by recurrent input (culture). Notably, total history dependence and the information timescale varied independently among neural systems, and studying them in isolation would miss differences, whereas considering them jointly allows to distinguish the different systems. Moreover, no clear differentiation between cortical culture, retina and primary visual cortex is possible using the autocorrelation time $\tau_C$ (Fig 7B), with medians $\tau_C \approx 68$ (culture), $\tau_C \approx 60$ (retina) and $\tau_C \approx 80$ (primary visual cortex), respectively.

To better understand how other well-established statistical measures relate to the total history dependence $R_{\text{tot}}$ and the information timescale $\tau_R$, we show $R_{\text{tot}}$ and $\tau_R$ versus the median interspike interval (ISI), the coefficient of variation $C_V = \sigma_{\text{ISI}}/\mu_{\text{ISI}}$ of the ISI distribution, and

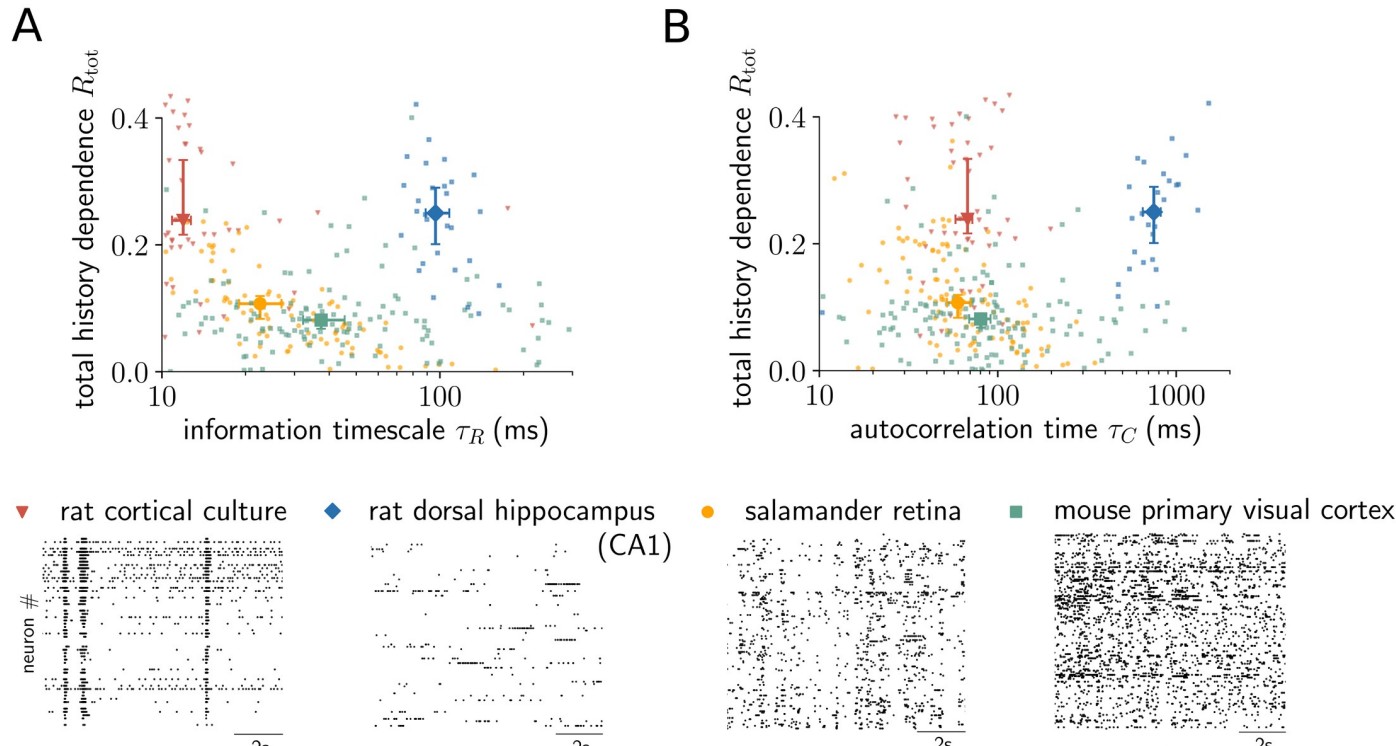

**Fig 7. Together, total history dependence and the information timescale show clear differences between neural systems.** (A) Embedding-optimized Shuffling estimates ($d_{max} = 5$) of the total history dependence $R_{tot}$ are plotted against the information timescale $\tau_R$ for individual sorted units (dots) from four different neural systems (raster plots show spike trains of different sorted units). No clear relationship between the two quantities is visible. The analysis shows systematic differences between the systems: sorted units in rat cortical culture ($n = 48$) and rat dorsal hippocampus layer CA1 ($n = 28$) have higher median total history dependence than units in salamander retina ($n = 111$) and mouse primary visual cortex ($n = 142$). At the same time, sorted units in cortical culture and retina show smaller timescale than units in primary visual cortex, and much smaller timescale than units in hippocampus layer CA1. Overall, neural systems are clearly distinguishable when jointly considering the total history dependence and information timescale. (B) Total history dependence $R_{tot}$ versus the autocorrelation time $\tau_C$ shows no clear relation between the two quantities, similar to the information timescale $\tau_R$. Also, the autocorrelation time gives the same relation in timescale between retina, primary visual cortex and CA1, whereas the cortical culture has a higher timescale (different order of medians on the x-axis). In general, neural systems are harder to differentiate in terms of the autocorrelation time $\tau_C$ compared to $\tau_R$. Errorbars indicate median over sorted units and 95% bootstrap confidence intervals on the median.

the autocorrelation time $\tau_C$ in S14 Fig. Estimates of the total history dependence $R_{tot}$ tend to decrease with the median ISI, and to increase with the coefficient of variation $C_V$. This result is expected for a measure of history dependence, because a shorter median ISI indicates that spikes tend to occur together, and a higher $C_V$ indicates a deviation from independent Poisson spiking. In contrast, the information timescale $\tau_R$ tends to increase with the autocorrelation time, as expected, with no clear relation to the median ISI or the coefficient of variation $C_V$. However, the correlation between the measures depends on the neural system. For example in retina ($n = 111$), $R_{tot}$ is significantly anti-correlated with the median ISI (Pearson correlation coefficient: $r = -0.69$, $p < 10^{-5}$) and strongly correlated with the coefficient of variation $C_V$ ($r = 0.90$, $p < 10^{-5}$), and $\tau_R$ is significantly correlated with the autocorrelation time $\tau_C$ ($r = 0.75$, $p < 10^{-5}$). In contrast, for mouse primary visual cortex ($n = 142$), we found no significant correlations between any of these measures. Thus, the relation between $R_{tot}$ or $\tau_R$ and the established measures is not systematic, and therefore one cannot replace the history dependence by any of them.

In addition to differences between neural systems, we also find strong heterogeneity of history dependence *within* a single system. Here, we demonstrate this for three different sorted units in primary visual cortex (Fig 8, see S9 Fig for all analyzed sorted units in primary visual

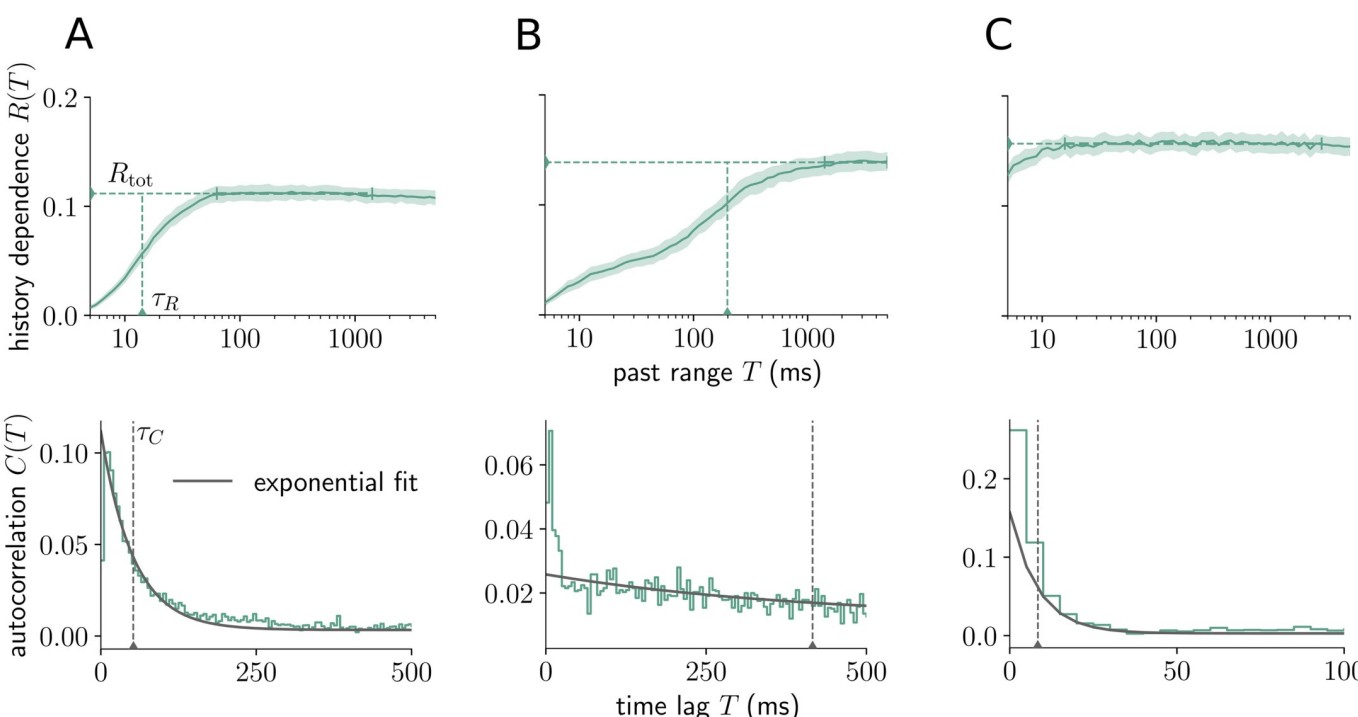

**Fig 8. Distinct signatures of history dependence for different sorted units within mouse primary visual cortex.** (Top) Embedding-optimized estimates of $R(T)$ reveal distinct signatures of history dependence for three different sorted units (A,B,C) within a single neural system (mouse primary visual cortex). In particular, sorted units have similar total history dependence $R_{\text{tot}}$, but differ vastly in the information timescale $\tau_R$ (horizontal and vertical dashed lines). Note that for unit C, $\tau_R$ is smaller than 5 ms and thus doesn't appear in the plot. Shaded areas indicate $\pm$ two standard deviations obtained by bootstrapping, and vertical bars indicate the interval over which estimates of $R(T)$ were averaged to estimate $R_{\text{tot}}$ (Materials and methods). Estimates were computed with the Shuffling estimator and $d_{\text{max}} = 5$. (Bottom) Autocorrelograms for the same sorted units (A,B, and C, respectively) roughly show an exponential decay, which was fitted (solid grey line) to estimate the autocorrelation time $\tau_C$ (grey dashed line). Similar to the information timescale $\tau_R$, only coefficients for $T$ larger than $T_0 = 10$ ms were considered during fitting.

cortex). In particular, sorted units display different signatures of history dependence $R(T)$ as a function of the past range $T$. For some units, history dependence builds up on short past ranges $T$ (e.g. Fig 8A), for some it only shows for higher $T$ (e.g. Fig 8B), and for some it already saturates for very short $T$ (e.g. Fig 8C). A similar behavior is captured by the autocorrelation $C(T)$ (Fig 8, second row). The rapid saturation in Fig 8C indicates history dependence due to bursty firing, which can also be seen by strong positive correlation with past spikes for short $T$ (Fig 8C, bottom). To exclude the effects of different firing modes or refractoriness on the information timescale, we only considered past ranges $T > T_0 = 10$ ms when estimating $\tau_R$, or time lags $T > T_0 = 10$ ms when fitting an exponential decay to $C(T)$ to estimate $\tau_C$. The reason is that differences in the integration of past information are expected to show for larger $T$. This agrees with the observation that timescales among neural systems were much more similar if one instead sets $T_0 = 0$ ms, whereas they showed clear differences for $T_0 = 10$ ms or $T_0 = 20$ ms (S15 Fig).

We also observed that history dependence can build up on all timescales up to seconds, and that it shows characteristic increases at particular past ranges, e.g. $T \approx 100$ ms and $T \approx 200$ ms in CA1 (Fig 6B), possibly reflecting phase information in the theta cycles [45, 46]. Thus, the analysis does not only serve to investigate differences in history dependence between neural systems, but also resolves clear differences between sorted units. This could be used to investigate differences in information processing between different cortical layers, different neuron types or neurons with different receptive field properties. Importantly, because units are so

**1) Embedding optimization:** The embedding of past spiking activity should be individually optimized to each spike train, in order to account for very different spiking statistics. This also applies to other information metrics like transfer entropy [52].

**2) Regularization:** Estimates have to be reliable lower bounds, otherwise one cannot interpret the results (apply Bayesian bias criterion or Shuffling correction).

**3) Exponential embedding:** Given the limitations on the number of bins, a non-uniform embedding is required to capture long-lasting dependencies. An exponential embedding with max. 5 bins is typically a good compromise between accuracy and computation speed, and enables embedding optimization for large, highly parallel spike recordings.

**4) Data requirements:** For practical purpose, spike recordings should be sufficiently long (at least 10 minutes). If several recordings are to be analyzed, these should be of similar length to allow for a meaningful comparison of history dependence and its timescale between recordings.

**Fig 9. Practical guidelines for the estimation of history dependence in single neuron spiking activity.** More details regarding the individual points can be found at the end of Materials and methods.

different, ad hoc embedding schemes with a fixed number of bins or fixed bin width will miss considerable history dependence.

## Discussion

To estimate history dependence in experimental data, we developed a method where the embedding of past spiking is optimized for each individual spike train. Thereby, it can estimate a maximum of history dependence, given what is possible for the limited amount of data. We found that embedding optimization is a robust and flexible estimation tool for neural spike trains with vastly different spiking statistics, where ad hoc embedding strategies would estimate substantially less history dependence. Based on our results, we arrived at practical guidelines that are summarized in Fig 9. In the following, we contrast history dependence $R(T)$ with time-lagged measures such as the autocorrelation in more detail, clearly discussing the advantages—but also the limitations of the approach. We then discuss how one can relate estimated history dependence to neural coding and information processing based on the example data sets analyzed in this paper.

### Advantages and limitations of history dependence in comparison to the autocorrelation and lagged mutual information

A key difference between history dependence $R(T)$ and the autocorrelation or lagged mutual information is that $R(T)$ quantifies statistical dependencies between current spiking and the *entire past spiking* in a past range $T$ (Fig 1B). This has the following advantages as a measure of statistical dependence, and as a footprint of information processing in single neuron spiking. First, $R(T)$ allows to compute the total history dependence, which, from a coding perspective, represents the redundancy of neural spiking with all past spikes; or how much of the past information is also represented when emitting a spike. Second, because past spikes are considered jointly, $R(T)$ captures synergistic effects and dismisses redundant past information (Fig 4). Finally, we found that this enables $R(T)$ to disentangle the strength and timescale of history dependence for the binary autoregressive process (Fig 3). In contrast, autocorrelation $C(T)$ or lagged mutual information $L(T)$ quantify the statistical dependence of neural spiking on a single past bin with time lag $T$, without considering any of the other bins (Fig 1A). Thereby, they miss synergistic effects; and they quantify redundant past dependencies that vanish once spiking activity in more-recent past is taken into account (Fig 4). As a consequence, the timescales

of these measures reflect both, the strength and the temporal depth of history dependence in the binary autoregressive process (Fig 3).

Moreover, technically, the autocorrelation time $\tau_C$ depends on fitting an exponential decay to coefficients $C(T)$. Computing the autocorrelation time with the generalized timescale is difficult, because coefficients $C(T)$ can be negative, and are too noisy for large $T$. While model fitting is in general more data efficient than the model-free estimation presented here, it can also produce biased and unreliable estimates [16]. Furthermore, when the coefficients do not decay exponentially, a more complex model has to be fitted [52], or the analysis simply cannot be applied. In contrast, the generalized timescale can be directly applied to estimates of the history dependence $R(T)$ to yield the information timescale $\tau_R$ without any further assumptions or fitting models. However, we found that estimates of $\tau_R$ can depend strongly on the estimation method and embedding dimension (S12 Fig) and the size of the data set (S2 and S3 Figs). The dependence on data size is less strong for the practical approach of optimizing up to $d_{\max} = 5$ past bins, but still we recommend to use data sets of similar length when aiming for comparability across experiments. Moreover, there might be cases where a model-free estimation of the true timescale might be infeasible because of the complexity of past dependencies (S2 Fig, neuron with a 22 seconds past kernel). In this case, only $\approx 80\%$ of the true timescale could be estimated on a 90 minute recording.

Another downside of quantifying the history dependence $R(T)$ is that its estimation requires more data than fitting the autocorrelation time $\tau_C$. To make best use of the limited data, we here devised the embedding optimization approach that allows to find the most efficient representation of past spiking for the estimation of history dependence. Even so, we found empirically that a minimum of 10 minutes of recorded spiking activity are advisable to achieve a meaningful quantification of history dependence and its timescale (S2 and S3 Figs). In addition, for shorter recordings, the analysis can lead to mild overestimation due to optimizing and overfitting embedding parameters on noisy estimates (S2 Fig). This overestimation can, however, be avoided by cross-validation, which we find to be particularly relevant for the Bayesian bias criterion (BBC) estimator. Finally, our approach uses an embedding model that ranges from uniform embedding to an embedding with exponentially stretching past bins—assuming that past information farther into the past requires less temporal resolution [53]. This embedding model might be inappropriate if, for example, spiking depends on the exact timing of distant past spikes, with gaps in time where past spikes are irrelevant. In such a case, embedding optimization could be used to optimize more complex embedding models that can also account for this kind of spiking statistics.

## Differences in total history dependence and information timescale between data sets agree with ideas from neural coding and hierarchical information processing

First, we found that the total history dependence $R_{\text{tot}}$ clearly differs among the experimental data sets. Notably, $R_{\text{tot}}$ was low for recordings of early visual processing areas such as retina and primary visual cortex, which is in line with the theory of efficient coding [1, 54] and neural adaptation for temporal whitening as observed in experiments [3, 55]. In contrast, $R_{\text{tot}}$ was high for neurons in dorsal hippocampus (layer CA1) and cortical culture. In CA1, the original study [46] found that the temporal structure of neural activity within the temporal windows set by the theta cycles was beyond of what one would expect from integration of feed-forward excitatory inputs. The authors concluded that this could be due to local circuit computations. The high values of $R_{\text{tot}}$ support this idea, and suggest that local circuit computations could serve the integration of past information, either for the formation of a path integration–based

neural map [56], or to recognize statistical structure for associative learning [8]. In cortical culture, neurons are exclusively driven by recurrent input and exhibit strong bursts in the population activity [57]. This leads to strong history dependence also at the single-neuron level.

To summarize, history dependence was low for early sensory processing and high for high level processing or past dependencies that are induced by strong recurrent feedback in a neural network. We thus conclude that estimated total history dependence $R_{tot}$ does indeed provide a footprint of neural coding and information processing.

Second, we observed that the information timescale $\tau_R$ increases from retina ($\approx 23$ ms) to primary visual cortex ($\approx 37$ ms) to CA1 ($\approx 96$ ms), in agreement with the idea of a temporal hierarchy in neural information processing [12]. These results qualitatively agree with similar results obtained for the autocorrelation time of spontaneous activity [9], although the information timescales are overall much smaller than the autocorrelation times. Our results suggest that the hierarchy of intrinsic timescales could also show in the history dependence of single neurons measured by the mutual information.

## Conclusion

Embedding optimization enables to estimate history dependence in a diversity of spiking neural systems, both in terms of its strength, as well as its timescale. The approach could be used in future experimental studies to quantify history dependence across a diversity of brain areas, e.g. using the novel Neuropixels probe [58], or even across cortical layers within a single area. To this end we provide a toolbox for Python3 [37]. These analyses might yield a more complete picture of hierarchical processing in terms of the timescale *and* a footprint of information processing and coding principles, i.e. information integration versus redundancy reduction.

## Materials and methods

In this section, we provide all mathematical details required to reproduce the results of this paper. We first provide the basic definitions of history dependence, time-lagged measures and the past embedding. We then describe the embedding optimization approach that is used to estimate history dependence from neural spike recordings, and provide a description of the workflow. Next, we delineate the estimators of history dependence considered in this paper, and present the novel Bayesian bias criterion. Finally, we provide details on the benchmark model and how we approximated its history dependence for given past range and embedding parameters. All code for Python3 that was used to analyze the data and to generate the figures is available online at https://github.com/Priesemann-Group/historydependence.

### Ethics statement

Data from salamander retina were recorded in strict accordance with the recommendations in the Guide for the Care and Use of Laboratory Animals of the National Institutes of Health, and the protocol was approved by the Institutional Animal Care and Use Committee (IACUC) of Princeton University (Protocol Number: 1828). The rat dorsal hippocampus experimental protocols were approved by the Institutional Animal Care and Use Committee of Rutgers University. Data from mouse primary visual cortex were recorded according to the UK Animals Scientific Procedures Act (1986).

### Glossary

**Terms**

- *Past embedding*: discrete, reduced representation of past spiking through temporal binning

- *Past-embedding optimization*: Optimization of temporal binning for better estimation of history dependence

- *Embedding-optimized estimate*: Estimate of history dependence for optimized embedding

  **Abbreviations**

- *GLM*: generalized linear model

- *ML*: Maximum likelihood

- *BBC*: Bayesian bias criterion

- *Shuffling*: Shuffling estimator based on a bias correction for the ML estimator

  **Symbols**

- $\Delta t$: bin size of the time bin for current spiking

- $T$: past range of the past embedding

- $[t - T, t)$: embedded past window

- $d$: embedding dimension or number of bins

- $\kappa$: scaling exponent for exponential embedding

- $T_{\mathrm{rec}}$: recording length

- $N = (T_{\mathrm{rec}} - T)/\Delta t$: number of measurements, i.e. number of observed joint events of current and past spiking

- $X$: random variable with binary outcomes $x \in [0, 1]$, which indicate the presence of a spike in a time bin $\Delta t$

- $X^{-T}$: random variable whose outcomes are binary sequences $\boldsymbol{X}^{-T} \in \{0, 1\}^d$, which represent past spiking activity in a past range $T$

  **Information-theoretic quantities**

- $H(\text{spiking}) \equiv H(X)$: average spiking information

- $H(\text{spiking}|\text{past}(T)) \equiv H(X|\boldsymbol{X}^{-T})$: average spiking information for given past spiking in a past range $T$

- $I(\text{spiking};\text{past}(T)) \equiv I(X; \boldsymbol{X}^{-T})$: mutual information between current spiking and past spiking in a past range $T$

- $R(T) \equiv I(X;\boldsymbol{X}^{-T})/H(X)$: history dependence for given past range $T$

- $R(T, d, \kappa) \equiv I(X; \boldsymbol{X}_{d,\kappa}^{-T})/H(X)$: history dependence for given past range $T$ and past embedding $d, \kappa$

- $R_{\mathrm{tot}} \equiv \lim\limits_{T \to \infty} R(T)$: total history dependence

- $\Delta R(T_i) \equiv R(T_i) - R(T_{i-1})$: gain in history dependence

- $\tau_R$: information timescale or generalized timescale of history dependence $R(T)$

- $L(T) \equiv I(X; X_{-T})$: lagged mutual information with time lag $T$

- $\tau_L$: generalized timescale of lagged mutual information $L(T)$

**Estimated quantities**

- $\hat{R}(T, d, \kappa)$: estimated history dependence for given past range $T$ and past embedding $d$, $\kappa$

- $\hat{R}(T)$: embedding-optimized estimate of $R(T)$ for optimal embedding parameters $d^*$, $\kappa^*$

- $\hat{R}_{\mathrm{tot}}$: estimated total history dependence, i.e. average $\hat{R}(T)$ for $T \in [T_D, T_{\max}]$, with interval of saturated estimates $[T_D, T_{\max}]$

- $\hat{\tau}_R$: estimated information timescale

## Basic definitions

**Definition of history dependence.**   We quantify history dependence $R(T)$ as the mutual information $I(X, \boldsymbol{X}^{-T})$ between present and past spiking $X$ and $\boldsymbol{X}^{-T}$, normalized by the binary Shannon information of spiking $H(X)$, i.e.

$$R(T) \equiv \frac{I(X, \boldsymbol{X}^{-T})}{H(X)} = 1 - \frac{H(X|\boldsymbol{X}^{-T})}{H(X)}. \tag{6}$$

Under the assumption of stationarity and ergodicity the mutual information can be computed either as the average over the stationary distribution $p(x, \boldsymbol{x}^{-T})$, or the time average [21, 58], i.e.

$$
\begin{aligned}
I(X, \mathbf{X}^{-T}) \quad &= H(X) - H(X|\mathbf{X}^{-T}) & (7) \\
&= \sum_{x \in \{0,1\}} p(x) \log_2 \frac{1}{p(x)} - \sum_{x \in \{0,1\}} \sum_{\mathbf{x}^{-T} \in \{0,1\}^d} p(x, \mathbf{x}^{-T}) \log_2 \frac{1}{p(x|\mathbf{x}^{-T})} & (8) \\
&= \sum_{x \in \{0,1\}} \sum_{\mathbf{x}^{-T} \in \{0,1\}^d} p(x, \mathbf{x}^{-T}) \log_2 \frac{p(x|\mathbf{x}^{-T})}{p(x)} & (9) \\
&= \lim_{N \to \infty} \frac{1}{N} \sum_{n=1}^{N} \log_2 \frac{p(x_{t_n}|\mathbf{x}_{t_n}^{-T})}{p(x_{t_n})}. & (10)
\end{aligned}
$$

Here, $x_{t_n} \in \{0, 1\}$ indicates the presence of a spike in a small interval $[t_n, t_n + \Delta t)$ with $\Delta t = 5$ ms throughout the paper, and $\boldsymbol{x}_{t_n}^{-T}$ encodes the spiking history in a time window $[t_n - T, t_n)$ at times $t_n = n\Delta t$ that are shifted by $\Delta t$.

**Definition of lagged mutual information.**   The lagged mutual information $L(T)$ [41] for a stationary neural spike trains is defined as the mutual information between present spiking $X$ and past spiking $X_{-T}$ with time lag $T$, i.e.

$$L(T) \equiv I(X; X_{-T}) \tag{11}$$

$$= \sum_{x \in \{0,1\}} \sum_{x_{-T} \in \{0,1\}} p(x, x_{-T}) \log_2 \frac{p(x|x_{-T})}{p(x)} \tag{12}$$

$$= \lim_{N \to \infty} \frac{1}{N} \sum_{n=1}^{N} \log_2 \frac{p(x_{t_n}|x_{t_n-T})}{p(x_{t_n})}. \tag{13}$$

Here, $x_{t_n} \in \{0, 1\}$ indicates the presence of a spike in a time bin $[t_n, t_n + \Delta t)$ and $x_{t_n-T} \in \{0, 1\}$ the presence of a spike in a single past bin $[t_n - T, t_n - T + \Delta t)$ at times $t_n = n\Delta t$ that are shifted by $\Delta t$. In analogy to $R(T)$, one can apply the generalized timescale to the lagged mutual

information to obtain a timescale $\tau_L$ with

$$\tau_L \equiv \sum_{i=1}^{n} \bar{T}_i \frac{L(T_i)}{\sum_{j=1}^{n} L(T_j)} - T_0. \tag{14}$$

**Definition of autocorrelation.** The autocorrelation $C(T)$ for a stationary neural spike train is defined as

$$C(T) = \frac{\mathrm{Cov}[x_{t_n}, x_{t_n-T}]}{\mathrm{Var}[x_{t_n}]} = \frac{\langle x_{t_n} x_{t_n-T} \rangle - \langle x_{t_n} \rangle^2}{\langle x_{t_n}^2 \rangle - \langle x_{t_n} \rangle^2} \tag{15}$$

with time lag $T$ and $x_{t_n}$ and $x_{t_n-T}$ as above. For an exponentially decaying autocorrelation $C(T) \propto \exp\left(-\frac{T}{\tau_C}\right)$, $\tau_C$ is called *autocorrelation time*.

**Past embedding.** Here, we encode the spiking history in a finite time window $[t - T, t)$ as a binary sequence $\boldsymbol{x}_t^{-T} = (x_{t,i}^{-T})_{i=1}^{d}$ of binary spike counts $x_{t,i}^{-T} \in \{0, 1\}$ in $d$ past bins (Fig 2). When more than one spike can occur in a single bin, $x_{t,i}^{-T} = 1$ is chosen for spike counts larger than the median activity in the $i$th bin. This type of temporal binning is more generally referred to as *past embedding*. It is formally defined as a mapping

$$\Gamma_T(\theta) : \mathcal{F}_T \to S^d \tag{16}$$

from the set of all possible spiking histories $\mathcal{F}_T = \sigma(\mathcal{X}_\tau : \tau \in [t - T, t))$, i.e. the sigma algebra generated by the point process $\mathcal{X}$ (neural spiking) in the time interval $[t - T, t)$, to the set of $d$-dimensional binary sequences $S^d$. We can drop the dependence on the time $t$ because we assume stationarity of the point process. Here, $T$ is the embedded *past range*, $d$ the *embedding dimension*, and $\theta$ denotes all the embedding parameters that govern the mapping, i.e. $\theta = (d, \ldots)$. The resulting binary sequence at time $t$ for given embedding $\theta$ and past range $T$ will be denoted by $\boldsymbol{x}_{t,\theta}^{-T}$. In this paper, we consider the following two embeddings for the estimation of history dependence.

**Uniform embedding.** If all bins have the same bin width $\tau = T/d$, the embedding is called *uniform*. The main drawback of the uniform embedding is that higher past ranges $T$ enforce a uniform decrease in resolution when $d$ is fixed.

**Exponential embedding.** One can generalize the uniform embedding by letting bin widths increase exponentially with bin index $j = 1, \ldots, d$ according to $\tau_j = \tau_1 10^{(j-1)\kappa}$. Here, $\tau_1$ gives the bin size of the first past bin, and is uniquely determined when $T$, $d$ and $\kappa$ are specified. Note that $\kappa = 0$ yields a uniform embedding, whereas $\kappa > 0$ decreases resolution on distant past spikes. For fixed embedding dimension $d$ and past range $T$, this allows to retain a higher resolution on spikes in the more-recent past.

**Sufficient embedding.** Ideally, the past embedding preserves all the information that the spiking history in the past range $T$ has about the present spiking dynamics. In that case, no additional past information has an influence on the probability for $x_t$ once the embedded spiking history $\boldsymbol{x}_{t,\theta}^{-T}$ is given, i.e.

$$p(x_t | \boldsymbol{x}_{t,\theta}^{-T}, \boldsymbol{x}_{t,v}^{-T}) = p(x_t | \boldsymbol{x}_{t,\theta}^{-T}) \tag{17}$$

for any other past embedding $\boldsymbol{x}_{t,v}^{-T}$. If Eq (17) holds for all times $t$, the embedding $\Gamma_T(\theta)$ is called a *sufficient* embedding. For the remainder of this paper, the sequences of sufficient embeddings are denoted by $\boldsymbol{x}_t^{-T}$.

**Insufficient embeddings cause underestimation of history dependence.** The past embedding is essential when inferring history dependence from recordings, because an insufficient embedding causes underestimation of history dependence. To show this, we note that for any embedding parameters $\theta$ and past range $T$ the Kullback-Leibler divergence between the spiking probability for the sufficient embedding $p(x_t|\boldsymbol{x}_t^{-T})$ and $p(x_t|\boldsymbol{x}_{t,\theta}^{-T})$ cannot be negative [60], i.e.

$$D_{KL}\left[p(x_t|\boldsymbol{x}_t^{-T})\|p(x_t|\boldsymbol{x}_{t,\theta}^{-T})\right] = \sum_{x_t \in \{0,1\}} p(x_t|\boldsymbol{x}_t^{-T}) \log_2 \frac{p(x_t|\boldsymbol{x}_t^{-T})}{p(x_t|\boldsymbol{x}_{t,\theta}^{-T})} \geq 0, \tag{18}$$

with equality *iff* $p(x_t|\boldsymbol{x}_{t,\theta}^{-T}) = p(x_t|\boldsymbol{x}_t^{-T})$. By taking the average over all times $t_n$, we arrive at

$$0 \leq \lim_{N\to\infty} \frac{1}{N} \sum_{n=1}^{N} \sum_{x_{t_n} \in \{0,1\}} p(x_{t_n}|\boldsymbol{x}_{t_n}^{-T}) \log_2 \frac{p(x_{t_n}|\boldsymbol{x}_{t_n}^{-T})}{p(x_{t_n}|\boldsymbol{x}_{t_n,\theta}^{-T})} \tag{19}$$

$$= \lim_{N\to\infty} \frac{1}{N} \sum_{n=1}^{N} \sum_{x_{t_n} \in \{0,1\}} p(x_{t_n}|\boldsymbol{x}_{t_n}^{-T}, \boldsymbol{x}_{t_n,\theta}^{-T}) \log_2 \frac{1}{p(x_{t_n}|\boldsymbol{x}_{t_n,\theta}^{-T})} \tag{20}$$

$$- \lim_{N\to\infty} \frac{1}{N} \sum_{n=1}^{N} \sum_{x_{t_n} \in \{0,1\}} p(x_{t_n}|\boldsymbol{x}_{t_n}^{-T}) \log_2 \frac{1}{p(x_{t_n}|\boldsymbol{x}_{t_n}^{-T})} \tag{21}$$

$$= H(X|\boldsymbol{X}_\theta^{-T}) - H(X|\boldsymbol{X}^{-T}), \tag{22}$$

where the last step follows from stationarity and ergodicity and marginalizing out $\boldsymbol{x}_{t_n}^{-T}$ in the first term. From here, it follows that one always underestimates the history dependence in neural spiking, as long as the embedding is not sufficient, i.e.

$$R(T, \theta) \equiv 1 - \frac{H(X|\boldsymbol{X}_\theta^{-T})}{H(X)} \leq 1 - \frac{H(X|\boldsymbol{X}^{-T})}{H(X)} = R(T). \tag{23}$$

## Estimation of history dependence using past-embedding optimization

The past embedding is crucial in determining how much history dependence we can capture, since an insufficient embedding $\theta$ leads to an underestimation of the history dependence $R(T) \geq R(T, \theta)$. In order to capture as much history dependence as possible, the embedding $\theta$ should be chosen to maximize the estimated history dependence $R(T, \theta)$. Since the history dependence has to be estimated from data, we formulate the following embedding optimization procedure in terms of the estimated history dependence $\hat{R}(T, \theta)$.

**Embedding optimization.** For given $T$, find the optimal embedding $\theta^*$ that maximizes the estimated history dependence

$$\theta^* = \arg\max_\theta \hat{R}(T, \theta). \tag{24}$$

This yields an *embedding-optimized* estimate $\hat{R}(T) = \hat{R}(T, \theta^*)$ of the true history dependence $R(T)$.

**Requirements.** Embedding optimization can only give sensible results if the optimized estimates $\hat{R}(T, \theta)$ are guaranteed to be unbiased or a lower bound to the true $R(T, \theta)$. Otherwise, embeddings will be chosen that strongly overestimate history dependence. In this paper,

we therefore use two estimators, BBC and Shuffling, the former of which is designed to be unbiased, and the latter a lower bound to the true $R(T, \theta)$ (see below). In addition, embedding optimization works only if the estimation variance is sufficiently small. Otherwise, maximizing over variable estimates can lead to a mild overestimation. We found for a benchmark model that this overestimation was negligibly small for a recording length of 90 minutes for a model neuron with a 4 Hz average firing rate (S1 Fig). For smaller recording lengths, potential over-fitting can be avoided by cross-validation, i.e. optimizing embeddings on one half of the recording and computing embedding-optimized estimates on the other half.

**Implementation.** For the optimization, we compute estimates $\hat{R}(T, d, \kappa)$ for a range of embedding dimensions $d \in [1, 2, \ldots, d_{\max}]$ and scaling parameters $\kappa = [0, \ldots, \kappa_{\max}]$. For each $T$, we then choose the optimal parameter combination $d^*, \kappa^*$ for each $T$ that maximizes the estimated history dependence $\hat{R}(T, d, \kappa)$, and use $\hat{R}(T, d^*, \kappa^*)$ as the best estimate of $R(T)$.

**Estimation of total history dependence and the information timescale.** When estimating history dependence $R(T)$ from data, there are some adjustments required to estimate the total history dependence $R_{\text{tot}}$ and the information timescale $\tau_R$.

First, estimates $\hat{R}(T)$ are not guaranteed to converge for large past ranges $T$, but might decrease due to a reduced resolution of embeddings for higher $T$ (Fig 2D). Thus, we estimated an interval $[T_D, T_{\max}]$ for which estimates have converged. Here, the temporal depth $T_D$ and the upper bound $T_{\max}$ are the first and the last past ranges $T$ for which estimates $\hat{R}(T)$ are within one standard deviation of the highest estimate $\hat{R}_{\max}$, i.e. $\hat{R}(T) \geq \hat{R}_{\max} - \sigma_{\hat{R}_{\max}}$ (Fig 2D, vertical blue bars). The standard deviation was estimated by bootstrapping (see Bootstrap confidence intervals). From this interval, an estimate of the total history dependence $\hat{R}_{\text{tot}}$ is obtained by averaging $\hat{R}(T)$ over past ranges $T \in [T_D, T_{\max}]$ (Fig 2D, horizontal dashed blue line).

Second, noisy estimates $\hat{R}(T)$ are not guaranteed to be monotonously increasing, hence increments $\Delta\hat{R}(T)$ can be negative. Moreover, noisy estimates can lead to positive $\Delta\hat{R}(T)$ even though the true $R(T)$ has already converged to $R_{\text{tot}}$. This can have a huge effect on the estimated information timescale $\hat{\tau}_R$ if one simply uses these estimates in Eq (5). To avoid this, we use knowledge about the behavior of the true $R(T)$ when estimating $\Delta R(T)$. In particular, we set estimates $\hat{R}(T)$ equal to the largest previous estimate $\hat{R}(T')$ for $T' < T$ if they fall below it, and equal to $\hat{R}_{\text{tot}}$ if they are larger than $\hat{R}_{\text{tot}}$. This enforces that the estimated gain $\Delta\hat{R}(T) \geq 0$ is non-negative, and excludes spurious gain for high $T$ due to noisy estimates.

Finally, the information timescale $\tau_R$ can crucially depend on the choice of the minimum past range $T_0$ in the sum in Eq (5). A $T_0 > 0$ larger than zero allows to ignore short term effects on the history dependence such as the refractory period or different firing modes, which we found beneficial for resolving differences in the timescale among different neural systems (S15 Fig). In contrast, if the decay is truly exponential, then $\tau_R$ is independent of $T_0$. In this paper, we chose $T_0 = 10$ ms to exclude short term effects, while also not excluding too much past information.

**Workflow.** The estimation workflow using embedding optimization is summarized in Fig 10.

## Different estimators of history dependence

To estimate $R(T, \theta)$, one has to estimate the binary entropy of spiking $H(X)$ in a small time bin $\Delta t$, and the conditional entropy $H(X|\boldsymbol{X}_\theta^{-T})$ from data. The estimation of the binary entropy only requires the average firing probability $p(x = 1) = r\Delta t$ with

$$\hat{H}(X) = -r\Delta t \log_2 r\Delta t - (1 - r\Delta t) \log_2 (1 - r\Delta t), \tag{25}$$

**1)** Define embeddings for fixed past range $T$.

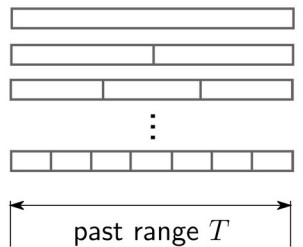

**2)** Record spike sequences for each embedding.

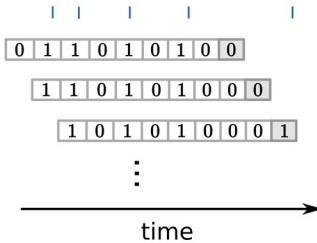

**3)** Estimate history dependence for each embedding.

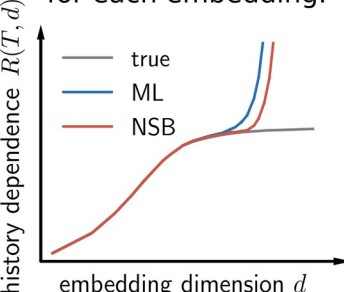

**4)** Apply regularization.

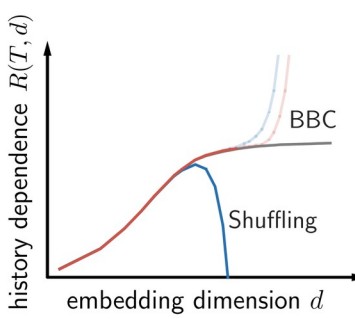

**5)** Select optimal embedding.

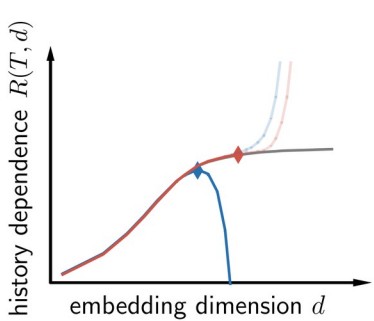

**6)** Repeat for all past ranges $T$ to estimate $R_{\text{tot}}$ and $\tau_R$.

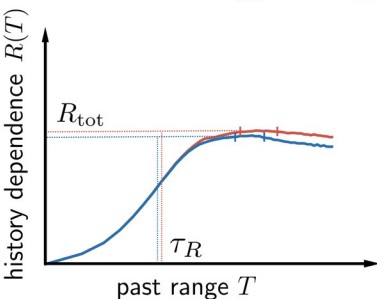

**Fig 10. Workflow of past-embedding optimization to estimate history dependence and the information timescale. 1)** Define a set of embedding parameters $d$, $\kappa$ for fixed past range $T$. **2)** For each embedding $d$, $\kappa$, record sequences of current and past spiking $x_{t_n}$, $\boldsymbol{x}_{t_n,\theta}^{-T}$ for all time steps $t_n$ in the recording. **3)** Use the frequencies of the recorded sequences to estimate history dependence for each embedding, either using maximum likelihood (ML), or fully Bayesian estimation (NSB). **4)** Apply regularization, i.e. the Bayesian bias criterion (BBC) or Shuffling bias correction, to ensure that all estimates are unbiased or lower bounds to the true history dependence. **5)** Select the optimal embedding to obtain an embedding-optimized estimate of $R(T)$. **6)** Repeat the estimation for a set of past ranges $T$ to compute estimates of the information timescale $\tau_R$ and the total history dependence $R_{\text{tot}}$.

which can be estimated with high accuracy from the estimated average firing rate $r$ even for short recordings. The conditional entropy $H(X|\boldsymbol{X}_\theta^{-T})$, on the other hand, is much more difficult to estimate. In this paper, we focus on a non-parametric approach that estimates

$$H(X|\boldsymbol{X}_\theta^{-T}) = H(X, \boldsymbol{X}_\theta^{-T}) - H(\boldsymbol{X}_\theta^{-T}) \tag{26}$$

by a non-parametric estimation of the entropies $H(\boldsymbol{X}_\theta^{-T})$ and $H(X, \boldsymbol{X}_\theta^{-T})$.

The estimation of entropy from data is a well-established problem, and we can make use of previously developed entropy estimation techniques for the estimation of history dependence. We here write out the estimation of the entropy term for joint sequences of present and past spiking $H(X, \boldsymbol{X}_\theta^{-T})$, which is the highest-dimensional term and thus the hardest to estimate. Estimation for the marginal entropy $H(\boldsymbol{X}_\theta^{-T})$ is completely analogous.

Computing the entropy requires knowing the statistical uncertainty and thus the probabilities for all possible joint sequences. In the following we will write probabilities as a vector $\boldsymbol{\pi} = (\pi_k)_{k=1}^K$, where $\pi_k \equiv p((x, \boldsymbol{x}_\theta^{-T}) = a_k)$ are the probabilities for the $K = 2^{d+1}$ possible joint spike patterns $a_k \in \{0, 1\}^{d+1}$. The entropy $H(X, \boldsymbol{X}_\theta^{-T})$ then reads

$$H(X, \boldsymbol{X}_\theta^{-T}) = H(\boldsymbol{\pi}) = -\sum_{k=1}^K \pi_k \log_2 \pi_k. \tag{27}$$

Once we are able to estimate the probability distribution $\boldsymbol{\pi}$, we are able to estimate the entropy.

In a non-parametric approach, the probabilities $\boldsymbol{\pi} = (\pi_k)_{k=1}^{K}$ are directly inferred from counts $\boldsymbol{n} = (n_k)_{k=1}^{K}$ of different spike sequences $a_k$ within the spike recording. Each time step $[t_n, t_n + \Delta t)$ provides a sample of present spiking $x_{t_n}$ and its history $\boldsymbol{x}_{t_n,\theta}^{-T}$, thus a recording of length $T_{\text{rec}}$ provides $N = (T_{\text{rec}} - T)/\Delta t$ data points.

**Maximum likelihood estimation.**   Most commonly, probabilities of spike sequences $a_k$ are then estimated as the relative frequencies $\hat{\pi}_k = n_k/N$ of their occurrence in the observed data. It is the maximum likelihood (ML) estimator of $\boldsymbol{\pi}$ for the multinomial likelihood

$$p(\boldsymbol{n}|\boldsymbol{\pi}) \propto \prod_{k=1}^{K} \pi_k^{n_k}. \tag{28}$$

Plugging the estimates $\hat{\pi}_k$ into the definition of the entropy in Eq (27) results in the ML estimator of the entropy

$$\hat{H}_{\text{ML}}(X, \boldsymbol{X}_{\theta}^{-T}) = -\sum_{k=1}^{K} \frac{n_k}{N} \log_2 \frac{n_k}{N} \tag{29}$$

or history dependence

$$\hat{R}_{\text{ML}}(T, \theta) = 1 - \frac{\hat{H}_{\text{ML}}(X, \boldsymbol{X}_{\theta}^{-T}) - \hat{H}_{\text{ML}}(\boldsymbol{X}_{\theta}^{-T})}{\hat{H}(X)}. \tag{30}$$

The ML estimator has the right asymptotic properties [28, 61], but is known to underestimate the entropy severely when data is limited [28, 62]. This is because all probability mass is assumed to be concentrated on the *observed* outcomes. A more concentrated probability distribution results in a smaller entropy, in particular if many outcomes have not been observed. This results in a systematic underestimation or negative bias

$$\text{Bias}[\hat{H}_{\text{ML}}(X, \boldsymbol{X}_{\theta}^{-T})] \leq 0. \tag{31}$$

The negative bias in the entropy, which is largest for the highest-dimensional joint entropy $\hat{H}_{\text{ML}}(X, \boldsymbol{X}_{\theta}^{-T})$, then typically leads to severe overestimation of the mutual information and history dependence [27, 63]. Because of this severe overestimation, we cannot use the ML estimator for embedding optimization.

**Bayesian Nemenman-Shafee-Bialek (NSB) estimator.**   In a Bayesian framework, the entropy is estimated as the posterior mean or minimum mean square error (MMSE)

$$\hat{H}_{\text{MMSE}}(\boldsymbol{n}) = \int d\boldsymbol{\pi} H(\boldsymbol{\pi}) p(\boldsymbol{\pi}|\boldsymbol{n}) = \int d\boldsymbol{\pi} H(\boldsymbol{\pi}) \frac{p(\boldsymbol{n}|\boldsymbol{\pi}) p(\boldsymbol{\pi})}{\int d\boldsymbol{\pi}' p(\boldsymbol{n}|\boldsymbol{\pi}') p(\boldsymbol{\pi}')}. \tag{32}$$

The posterior mean is the mean of the entropy with respect to the posterior distribution on the probability vector $\boldsymbol{\pi}$ given the observed frequencies of spike sequences $\boldsymbol{n}$

$$p(\boldsymbol{\pi}|\boldsymbol{n}) = \frac{p(\boldsymbol{n}|\boldsymbol{\pi}) p(\boldsymbol{\pi})}{\int d\boldsymbol{\pi}' p(\boldsymbol{n}|\boldsymbol{\pi}') p(\boldsymbol{\pi}')}. \tag{33}$$

The probability $p(\mathbf{n}|\boldsymbol{\pi})$ for i.i.d. observations $\boldsymbol{n}$ from an underlying distribution $\boldsymbol{\pi}$ is given by the multinomial distribution in Eq (28).

If the prior $p(\boldsymbol{\pi})$ is a conjugate prior to the multinomial likelihood, then the high-dimensional integral of Eq (32) can be evaluated analytically [32]. This is true for a class of priors

called Dirichlet priors, and in particular for symmetric Dirichlet priors

$$p(\boldsymbol{\pi}|\beta) \propto \prod_{k=1}^{K} \pi_k^{\beta-1}. \tag{34}$$

The prior $p(\boldsymbol{\pi}|\beta)$ gives every outcome the same a priori weight, but controls the weight $\beta > 0$ of uniform prior pseudo-counts. A $\beta = 1$ corresponds to a flat prior on all probability distributions $\boldsymbol{\pi}$, whereas $\beta \to 0$ gives maximum likelihood estimation (no prior pseudo-count).

It has been shown that the choice of $\beta$ is highly informative with respect to the entropy, in particular when the number of outcomes $K$ becomes large [64]. This is because the a priori variance of the entropy vanishes for $K \to \infty$, thus for any $\boldsymbol{\pi} \sim p(\boldsymbol{\pi}|\beta)$ the entropy $H(\boldsymbol{\pi})$ is very close to the a priori expected entropy

$$\xi(\beta) = \int d\boldsymbol{\pi} H(\boldsymbol{\pi}) p(\boldsymbol{\pi}|\beta) = \psi_0(K\beta + 1) - \psi_0(\beta + 1), \tag{35}$$

where $\psi_m(z) = \partial_z^{m+1} \log \Gamma(z)$ are the polygamma functions. In addition, a lot of data is required to counter-balance this a priori expectation. This is because the prior adds pseudo-counts on every outcome, i.e. it assumes that every outcome has been observed $\beta$ times prior to inference. In order to influence a prior that constitutes $K$ pseudo-counts, one needs at least $N > K$ samples, with more data required the sparser the true underlying distribution. Therefore, an estimator of the entropy for little data and fixed concentration parameter $\beta$ is highly biased towards the a priori expected entropy $\xi(\beta)$.

Nemenman et al. [33] exploited the tight link between concentration parameter $\beta$ and the a priori expected entropy to derive a mixture prior

$$p_{NSB}(\boldsymbol{\pi}) \quad \propto \int d\beta \left| \frac{\partial \xi}{\partial \beta} \right| p(\boldsymbol{\pi}|\beta), \tag{36}$$

$$\frac{\partial \xi}{\partial \beta} \quad = K\psi_1(K\beta + 1) - \psi_1(\beta + 1), \tag{37}$$

that weights Dirichlet priors to be flat with respect to the expected entropy $\xi(\beta)$. Since the variance of this expectation vanishes for $K \gg 1$ [64], for high $K$ the prior is also approximately flat with respect to the entropy, i.e. $H(\boldsymbol{\pi}) \sim \mathcal{U}(0, \log_2 K)$ for $\boldsymbol{\pi} \sim p_{NSB}(\boldsymbol{\pi})$. The resulting MMSE estimator for the entropy is referred to as the NSB estimator

$$\hat{H}_{NSB}(\boldsymbol{n}) \quad = \int d\boldsymbol{\pi} H(\boldsymbol{\pi}) \frac{p(\boldsymbol{n}|\boldsymbol{\pi}) p_{NSB}(\boldsymbol{\pi})}{\int d\boldsymbol{\pi}' p(\boldsymbol{n}|\boldsymbol{\pi}') p_{NSB}(\boldsymbol{\pi}')} \tag{38}$$

$$= \frac{\int d\beta \frac{d\xi}{d\beta}(\beta) \hat{H}(\beta) \rho(\beta, \boldsymbol{n})}{\int d\beta' \frac{d\xi}{d\beta}(\beta') \rho(\beta', \boldsymbol{n})}. \tag{39}$$

Here, $\rho(\beta, \boldsymbol{n})$ is proportional to the evidence for given concentration parameter

$$\rho(\beta, \boldsymbol{n}) := \frac{\Gamma(K\beta)}{\Gamma(N + K\beta)} \prod_{i=1}^{K} \frac{\Gamma(n_i + \beta)}{\Gamma(\beta)} \tag{40}$$

$$\propto \int d\boldsymbol{\pi} \, p(\boldsymbol{n}|\boldsymbol{\pi}) \, p(\boldsymbol{\pi}|\beta) = p(\boldsymbol{n}|\beta), \tag{41}$$

where $\Gamma(x)$ is the gamma function. The posterior mean of the entropy for given concentration

parameter is

$$\hat{H}(\beta) = \sum_{i=1}^{K} \frac{n_i + \beta}{N + K\beta} [\psi_0(N + K\beta + 1) - \psi_0(n_i + \beta + 1)]. \tag{42}$$

From the Bayesian entropy estimate, we obtain an NSB estimator for history dependence

$$\hat{R}_{\text{NSB}}(T, \theta) = 1 - \frac{\hat{H}_{\text{NSB}}(X, \boldsymbol{X}_\theta^{-T}) - \hat{H}_{\text{NSB}}(\boldsymbol{X}_\theta^{-T})}{\hat{H}(X)}, \tag{43}$$

where the marginal and joint entropies are estimated individually using the NSB method.

To compute the NSB entropy estimator, one has to perform a one-dimensional integral over all possible concentration parameters $\beta$. This is crucial to be unbiased with respect to the entropy. An implementation of the NSB estimator for Python3 is published alongside the paper with our toolbox [37]. To compute the integral, we use a Gaussian approximation around the maximum a posteriori $\beta^*$ to define sensible integration bounds when the likelihood is highly peaked, as proposed in [34].

**Bayesian bias criterion.** The goal of the Bayesian bias criterion (BBC) is to indicate when estimates of history dependence are potentially biased. It might indicate bias even when estimates are unbiased, but the opposite should never be true.

To indicate a potential estimation bias, the BBC compares ML and NSB estimates of the history dependence. ML estimates are biased when too few joint sequences have been observed, such that the probability for unobserved or undersampled joint outcomes is underestimated. To counterbalance this effect, the NSB estimate adds $\beta$ pseudo-counts to every outcome, and then infers $\beta$ with an uninformative prior. For the BBC, we turn the idea around: when the assumption of no pseudo-counts (ML) versus a posterior belief on non-zero pseudo-counts (NSB) yield different estimates of history dependence, then too few sequences have been observed and estimates are potentially biased. This motivates the following definition of the BBC.

The NSB estimator $R_{\text{NSB}}(T, \theta)$ is biased with tolerance $p > 0$, if

$$|\hat{R}_{\text{NSB}}(T, \theta) - \hat{R}_{\text{ML}}(T, \theta)| > p \cdot \hat{R}_{\text{NSB}}(T, \theta). \tag{44}$$

Similarly, we define the BBC estimator

$$\hat{R}_{\text{BBC}}(T, \theta) \equiv \begin{cases} \hat{R}_{\text{NSB}}(T, \theta) & \text{if} \quad |\hat{R}_{\text{NSB}}(T, \theta) - \hat{R}_{\text{ML}}(T, \theta)| \leq p \cdot \hat{R}_{\text{NSB}}(T, \theta), \\ 0 & \text{otherwise.} \end{cases} \tag{45}$$

This estimator is designed to be unbiased, and thus can be used for embedding optimization in Eq (24). We use the NSB estimator for $R(T, \theta)$ instead of the ML estimator, because it is generally less biased. A tolerance $p > 0$ accounts for this, and accepts NSB estimates when there is only a small difference between the estimates. The bound for the difference is multiplied by $\hat{R}_{\text{NSB}}(T, \theta)$, because this provides the scale on which one should be sensitive to estimation bias. We found that a tolerance of $p = 0.05$ was small enough to avoid overestimation by BBC estimates on the benchmark model (Fig 5 and S1 Fig).

**Shuffling estimator.** The Shuffling estimator was originally proposed in [31] to reduce the sampling bias of the ML mutual information estimator. It has the desirable property that it is negatively biased in leading order of the inverse number of samples. Because of this property, Shuffling estimates can safely be maximized during embedding optimization without the

risk of overestimation. Here, we therefore propose to use the Shuffling estimator for embedding-optimized estimation of history dependence.

The idea behind the Shuffling estimator is to rewrite the ML estimator of history dependence as

$$\hat{R}_{\mathrm{ML}}(T, \theta) = \frac{1}{\hat{H}(X)} \left( \hat{H}_{\mathrm{ML}}(\boldsymbol{X}_\theta^{-T}) - \hat{H}_{\mathrm{ML}}(\boldsymbol{X}_\theta^{-T}|X) \right) \tag{46}$$

and to correct for bias in the entropy estimate $\hat{H}_{\mathrm{ML}}(\boldsymbol{X}_\theta^{-T}|X)$. Since $X$ is well sampled and thus $\hat{H}(X)$ is unbiased, and the bias of the ML entropy estimator is always negative [28, 61], we know that

$$\mathrm{Bias}[\hat{R}_{\mathrm{ML}}(T, \theta)] = \mathrm{Bias}[\hat{H}_{\mathrm{ML}}(\boldsymbol{X}_\theta^{-T})] - \mathrm{Bias}[\hat{H}_{\mathrm{ML}}(\boldsymbol{X}_\theta^{-T}|X)] \tag{47}$$

$$\leq -\mathrm{Bias}[\hat{H}_{\mathrm{ML}}(\boldsymbol{X}_\theta^{-T}|X)]. \tag{48}$$

Therefore, if we find a correction term of the magnitude of $\mathrm{Bias}[\hat{H}_{\mathrm{ML}}(\boldsymbol{X}_\theta^{-T}|X)]$, we can turn the bias in the estimate of the history dependence from positive to negative, thus obtaining an estimator that is a lower bound of the true history dependence. This can be achieved by subtracting a lower bound of the estimation bias $\mathrm{Bias}[\hat{H}_{\mathrm{ML}}(\boldsymbol{X}_\theta^{-T}|X)]$ from $\hat{H}_{\mathrm{ML}}(\boldsymbol{X}_\theta^{-T}|X)$.

In the following, we describe how [31] obtain a lower bound of the bias in the conditional entropy $\hat{H}_{\mathrm{ML}}(\boldsymbol{X}_\theta^{-T}|X)$ by computing the estimation bias for shuffled surrogate data.

Surrogate data are created by shuffling recorded spike sequences such that statistical dependencies between past bins are eliminated. This is achieved by taking all past sequences that were followed by a spike, and permuting past observations of the same bin index $j$. The same is repeated for all past sequences that were followed by no spike. The underlying probability distribution can then be computed as

$$p_{\mathrm{sh}}(\boldsymbol{x}_\theta^{-T}|x) = \prod_{j=1}^{d} p(x_{\theta,j}^{-T}|x), \tag{49}$$

and the corresponding entropy is

$$H(\boldsymbol{X}_{\theta,\mathrm{sh}}^{-T}|X) = \sum_{j=1}^{d} H(X_{\theta,j}^{-T}|X). \tag{50}$$

The pairwise probabilities $p(x_{\theta,j}^{-T}|x)$ are well sampled, and thus each conditional entropy in the sum can be estimated with high precision. This way, the true conditional entropy $H(\boldsymbol{X}_{\theta,\mathrm{sh}}^{-T}|X)$ for the shuffled surrogate data can be computed and compared to the ML estimate $\hat{H}_{\mathrm{ML}}(\boldsymbol{X}_{\theta,\mathrm{sh}}^{-T}|X)$ on the shuffled data. The difference between the two

$$\Delta\hat{H}_{\mathrm{ML}}(\boldsymbol{X}_{\theta,\mathrm{sh}}^{-T}|X)] \equiv \hat{H}_{\mathrm{ML}}(\boldsymbol{X}_{\theta,\mathrm{sh}}^{-T}|X) - H(\boldsymbol{X}_{\theta,\mathrm{sh}}^{-T}|X) \tag{51}$$

yields a correction term that is on average equal to the bias of the ML estimator on the shuffled data.

Importantly, the bias of the ML estimator on the shuffled data is in leading order more negative than on the original data. To see this, we consider an expansion of the bias on the

conditional entropy in inverse powers of the sample size $N$ [27, 63]

$$\text{Bias}[\hat{H}_{\text{ML}}(\boldsymbol{X}_\theta^{-T}|X)] = -\frac{1}{2N \ln 2} \sum_{x \in \{0,1\}} (\tilde{K}(x) - 1) + \mathcal{O}\left(\frac{1}{N^2}\right). \tag{52}$$

Here, $\tilde{K}(x)$ denotes the number of past sequences with nonzero probability $p(\boldsymbol{x}_\theta^{-T} = a_k|x) > 0$ of being observed when followed by a spike ($x = 1$) or no spike ($x = 0$), respectively. Notably, the bias is negative in leading order, and depends only on the number of possible sequences $\tilde{K}(x)$. For the shuffled surrogate data, we know that $p_{\text{sh}}(\boldsymbol{x}_\theta^{-T} = a_k|x) = 0$ implies $p(\boldsymbol{x}_\theta^{-T} = a_k|x) = 0$, but Shuffling may lead to novel sequences that have zero probability otherwise. Hence the number of possible sequences under Shuffling can only increase, i.e. $\tilde{K}_{\text{sh}}(x) \geq \tilde{K}(x)$, and thus the bias of the ML estimator under Shuffling to first order is always more negative than for the original data

$$\text{Bias}[\hat{H}_{\text{ML}}(\boldsymbol{X}_{\theta,\text{sh}}^{-T}|X)] \lesssim \text{Bias}[\hat{H}_{\text{ML}}(\boldsymbol{X}_\theta^{-T}|X)]. \tag{53}$$

Terms that could render it higher are of order $\mathcal{O}(N^{-2})$ and are assumed to have no practical relevance.

This motivates the following definition of the Shuffling estimator: Compute the difference between the ML estimator on the shuffled and original data to yield a bias-corrected Shuffling estimate

$$\hat{H}_{\text{ML,sh}}(\boldsymbol{X}_\theta^{-T}|X) \equiv \hat{H}_{\text{ML}}(\boldsymbol{X}_\theta^{-T}|X) - \Delta\hat{H}_{\text{ML}}(\boldsymbol{X}_{\theta,\text{sh}}^{-T}|X), \tag{54}$$

and use this to estimate history dependence

$$\hat{R}_{\text{Shuffling}}(T, \theta) \equiv \frac{1}{\hat{H}(X)} \left(\hat{H}_{\text{ML}}(\boldsymbol{X}_\theta^{-T}) - \hat{H}_{\text{ML,sh}}(\boldsymbol{X}_\theta^{-T}|X)\right). \tag{55}$$

Because of Eqs (48) and (53), we know that this estimator is negatively biased in leading order

$$\text{Bias}[\hat{R}_{\text{Shuffling}}(T, \theta)] \lesssim 0 \tag{56}$$

and can safely be used for embedding optimization.

**Estimation of history dependence by fitting a generalized linear model (GLM).**
Another approach to the estimation of history dependence is to model the dependence of neural spiking onto past spikes explicitly, and to fit model parameters to maximize the likelihood of the observed spiking activity [21]. For a given probability distribution $p(x_t|\boldsymbol{x}_t^{-T}, v)$ of the model with parameters $v$, the conditional entropy can be estimated as

$$\hat{H}(X|\boldsymbol{X}^{-T}, v) = \frac{1}{N} \sum_{n=1}^{N} \log_2 p(x_{t_n}|\boldsymbol{x}_{t_n}^{-T}, v)^{-1} \tag{57}$$

which one can plug into Eq (6) to obtain an estimate of the history dependence. The strong law of large numbers [59] ensures that if the model is correct, i.e. $p(x_t|\boldsymbol{x}_t^{-T}, v) = p(x_t|\boldsymbol{x}_t^{-T})$ for all $t$, this estimator converges to the entropy $H(X|\boldsymbol{X}^{-T})$ for $N \to \infty$. However, any deviations from the true distribution due to an incorrect model will lead to an underestimation of history dependence, similar to choosing an insufficient embedding. Therefore, model parameters should be chosen to maximize the history dependence, or to maximize the likelihood

$$v^* = \arg\max_v \sum_{n=1}^{N} \log_2 p(x_{t_n}|\boldsymbol{x}_{t_n}^{-T}, v). \tag{58}$$

We here consider a generalized linear model (GLM) with exponential link function that has successfully been applied to make predictions in neural spiking data [20] and can be used for the estimation of directed, causal information [21]. In a GLM with past dependencies, the spiking probability at time $t$ is described by the instantaneous rate or conditional intensity function

$$\lambda(t|\boldsymbol{x}_t^{-T}, v) = \lim_{\delta t \to 0} \frac{p(\hat{t} \in [t, t + \delta t]|\boldsymbol{x}_t^{-T}, v)}{\delta t}. \tag{59}$$

Since we discretize spiking activity in time as spiking or non-spiking in a small time window $\Delta t$, the spiking probability is given by the binomial probability

$$p(x_t = 1|\boldsymbol{x}_t^{-T}, v) = \frac{\lambda(t|\boldsymbol{x}_t^{-T}, v)\Delta t}{1 + \lambda(t|\boldsymbol{x}_t^{-T}, v)\Delta t}. \tag{60}$$

The idea of the GLM is that past events contribute independently to the probability of spiking, such that the conditional intensity function factorizes over their contributions. Hence, it can be written as

$$\lambda(t|\boldsymbol{x}_t^{-T}, \mu, \boldsymbol{h}) = \exp\left(\mu + \sum_{j=1}^{d} h_j x_{t,j}^{-T}\right), \tag{61}$$

where $h_j$ gives the contribution of past activity $x_{t,j}^{-T}$ in past time bin $j$ to the firing rate, and $\mu$ is an offset that is adapted to match the average firing rate.

Although fitting GLM parameters is more data-efficient than computing non-parametric estimates, overfitting may occur for limited data and high embedding dimensions $d$, hence $d$ cannot be chosen arbitrarily high. In order to estimate a maximum of history dependence for limited $d$, we apply the same type of binary past embedding as we use for the other estimators, and optimize the embedding parameters by minimizing the Bayesian information criterion [65]. In particular, for given past range $T$, we choose embedding parameters $d^*$, $\kappa^*$ that minimize

$$\text{BIC}(d, \kappa) = (d + 1) \log_2 N - 2\mathcal{L}^*(d, \kappa), \tag{62}$$

where $N$ is the number of samples and

$$\mathcal{L}^*(d, \kappa) = \sum_{n=1}^{N} \log_2 p(x_{t_n}|\boldsymbol{x}_{t_n, d, \kappa}^{-T}, \mu^*, \boldsymbol{h}^*) \tag{63}$$

is the maximized log-likelihood of the recorded spike sequences $(x_{t_n}, \boldsymbol{x}_{t_n, d, \kappa}^{-T})_{n=1}^{N}$ for optimal model parameters $\mu^*$, $\boldsymbol{h}^*$. We then use the optimized embedding parameters to estimate the conditional entropy according to

$$\hat{H}_{\text{GLM}}(X|\boldsymbol{X}_{d^*, \kappa^*}^{-T}) = -\frac{1}{N}\mathcal{L}^*(d^*, \kappa^*), \tag{64}$$

which results in the GLM estimator of history dependence

$$\hat{R}_{\text{GLM}}(T) = 1 - \frac{\hat{H}_{\text{GLM}}(X|\boldsymbol{X}_{d^*, \kappa^*}^{-T})}{\hat{H}(X)}. \tag{65}$$

**Bootstrap confidence intervals.** In order to estimate confidence intervals of estimates $\hat{R}(T, \theta)$ for given past embeddings, we apply the *blocks of blocks* bootstrap method [66]. To obtain bootstrap samples, we first compute all the binary sequences $(x_{t_n}, \boldsymbol{x}_{t_n, \theta}^{-T})$ for $n = 1, \ldots, N$ that result from discretizing the spike recording in $N$ time steps $\Delta t$ and applying the past embedding. We then randomly draw $N/l$ blocks of length $l$ of the recorded binary sequences such that the total number of redrawn sequences is the same as the in the original data. We choose $l$ to be the average interspike interval (ISI) in units of time steps $\Delta t$, i.e. $l = 1/(r\Delta t)$ with average firing rate $r$. Sampling successive sequences over the typical ISI ensures that bootstrap samples are representative of the original data, while also providing a high number of distinct blocks that can be drawn.

The different estimators (but not the bias criterion) are then applied to each bootstrap sample to obtain confidence intervals of the estimates. Instead of computing the 95% confidence interval via the 2.5 and 97.5 percentiles of the bootstrapped estimates, we assumed a Gaussian distribution and approximated the interval via $[\hat{R}(T, \theta) - 2\hat{\sigma}_R(T, \theta), \hat{R}(T, \theta) + 2\hat{\sigma}_R(T, \theta)]$, where $\hat{\sigma}_R(T, \theta)$ is the standard deviation over the bootstrapped estimates.

We found that the true standard deviation of estimates for the model neuron was well estimated by the bootstrapping procedure, irrespective of the recording length (S10 Fig). Furthermore, we simulated 100 recordings of the same recording length, and for each computed confidence interval for the past range $T$ with the highest estimated history dependence $R(T)$. By measuring how often the model's true value for the same embedding was included in these intervals, we found that the Gaussian confidence intervals are indeed close to the claimed confidence level (S10 Fig). This indicates that the bootstrap confidence intervals approximate well the uncertainty associated with estimates of history dependence.

**Cross-validation.** For small recording lengths, embedding optimization may cause overfitting through the maximization of variable estimates (S1 Fig). To avoid this type of overestimation, we apply one round of cross-validation, i.e. we optimize embeddings over the first half of the recording, and evaluate estimates for the optimal past embedding on the second half. We chose this separation of training and evaluation data sets, because it allows the fastest computation of binary sequences $(x_{t_n}, \boldsymbol{x}_{t_n, \theta}^{-T})$ for the different embeddings during optimization. We found that none of the cross-validated embedding-optimized estimates were systematically overestimating the true history dependence for the benchmark model for recordings as short as three minutes (S1 Fig). Therefore, cross-validation allows to apply embedding optimization to estimate history dependence even for short recordings.

## Benchmark neuron model

**Generalized leaky integrate-and-fire neuron with spike-frequency adaptation.** As a benchmark model, we chose a generalized leaky integrate-and-fire model (GLIF) with an additional adaptation filter $\xi$ (GLIF-$\xi$) that captures spike-frequency adaptation over 20 seconds [43].

For a standard leaky integrate-and-fire neuron, the neuron's membrane is formalized as an RC circuit, where the cell's lipid membrane is modeled as a capacitance $C$, and the ion channels as a resistance that admits a leak current with effective conductance $g_L$. Hence, the temporal evolution of the membrane's voltage $V$ is governed by

$$C\dot{V} = -g_L(V - V_R) + I_{\text{ext}}(t). \qquad (66)$$

Here, $V_R$ denotes the resting potential and $I_{\text{ext}}(t)$ external currents that are induced by some external drive. The neuron emits an action potential (spike) once the neuron crosses a voltage

threshold $V_T$, where a spike is described as a delta pulse at the time of emission $\hat{t}$. After spike emission, the neuron returns to a reset potential $V_0$. Here, we do not incorporate an explicit refractory period, because interspike intervals in the simulation were all larger than 10 ms. For constant input current $I_{\text{ext}}$, integrating Eq (66) yields the membrane potential between two spiking events

$$V(t) = V_\infty + (V_0 - V_\infty)e^{-\gamma(t-\hat{t}_0)}, \tag{67}$$

where $\hat{t}_0$ is the time of the most recent spike, $\gamma = g_L/C$ the inverse membrane timescale and $V_\infty = V_R + I_{\text{ext}}/\gamma$ the equilibrium potential.

In contrast to the LIF, the GLIF models the spike emission with a soft spiking threshold. To do that, spiking is described by an inhomogeneous Poisson process, where the spiking probability in a time window of width $\delta t \ll 1$ is given by

$$p(\hat{t} \in [t, t + \delta t]) = 1 - \exp\left(\int_t^{t+\delta t} \lambda(s)ds\right) \approx \lambda(t)\delta t. \tag{68}$$

Here, the spiking probability is governed by the time dependent firing rate

$$\lambda(t) = \lambda_0 \exp\left(\frac{V(t) - V_T(t)}{\Delta V}\right). \tag{69}$$

The idea is that once the membrane potential $V(t)$ approaches the firing threshold $V_T(t)$, the firing probability increases exponentially, where the exponential increase is modulated by $1/\Delta V$. For $\Delta V \to 0$, we recover the deterministic LIF, while for larger $\Delta V$ the emission becomes increasingly random.

In the GLIF-$\xi$, the otherwise constant threshold $V_T^*$ is modulated by the neuron's own past activity according to

$$V_T(t) = V_T^* + \sum_{\hat{t}_j < t} \xi(t - \hat{t}_j). \tag{70}$$

Thus, depending on their spike times $\hat{t}_j$, emitted action potentials increase or decrease the threshold additively and independently according to an adaptation filter $\xi(t)$. In the experiments conducted in [43], the following functional form for the adaptation filter was extracted:

$$\xi(s) = \begin{cases} a_\xi & , \text{ if } 0 < s \leq T_\xi \\ a_\xi\left(\frac{s}{T_\xi}\right)^{-\beta_\xi} & , \text{ if } T_\xi < s < 22\text{s}. \end{cases} \tag{71}$$

The filter is an effective model not only for the measured increase in firing threshold, but also for spike-triggered currents that reduce the membrane potential. When mapped to the effective adaptation filter $\xi$, it turned out that past spikes lead to a decrease in firing probability that is approximately constant over a period $T_\xi = 8.3$ ms, after which it decays like a power-law with exponent $\beta_\xi = 0.93$, until the contributions are set to zero after 22 s.

**Model variant with 1 s past kernel.** For demonstration, we also simulated a variant of the above model with a 1 s past kernel

$$\xi^{1s}(s) = \begin{cases} a_\xi^{1s} & , \text{if } 0 < s \leq T_\xi \\ a_\xi^{1s}\left(\frac{s}{T_\xi}\right)^{-\beta_\xi} & , \text{if } T_\xi < s < 1\text{ s}. \end{cases} \tag{72}$$

All parameters are identical apart from the strength of the kernel $a_\xi^{1s} = 35.2 \, \text{mV}$, which was adapted to maintain a firing rate of 4 Hz despite the shorter kernel.

**Simulation details.** In order to ensure stationarity, we simulated the model neuron exposed to a constant external current $I_{\text{ext}} = const.$ over a total duration of $T_{\text{rec}} = 900$ min. Thereby, the current $I_{\text{ext}}$ was chosen such that the neuron fired with a realistic average firing rate of 4 Hz. During the simulation, Eq (66) was integrated using simple Runge-Kutta integration with an integration time step of $\delta t = 0.5$ ms. At every time step, random spiking was modeled as a binary variable with probability as in Eq (68). After a burning-in time of 100 s, spike times were recorded and used for the estimation of history dependence. The detailed simulation parameters can be found in Table 1.

**Computation of the total history dependence.** In order to determine the total history dependence in the simulated spiking activity, we computed the conditional entropy $H(X|X^{-\infty})$ from the conditional spiking probability in Eq (68) that was used for the simulation. Note that this is only possible because of the constant input current, otherwise the conditional spiking probability would also capture information about the external input.

Since the conditional probability of spiking used in the simulation computes the probability in a simulation step $\delta t = 0.5$ ms, we first have to transform this to a probability of spiking in the analysis time step $\Delta t = 5$ ms. To do so, we compute the probability of no spike in a time step $[t, t + \Delta t)$ according to

$$p_{\text{sim}}(x_t = 0 | \mathbf{x}_t^{-\infty}) = \prod_{j=1}^{\Delta t/\delta t} [1 - \tilde{\lambda}(t + (j-1)\delta t)\delta t], \qquad (73)$$

and then compute the probability of at least one spike by $p(x_t = 1 | \mathbf{x}_t^{-\infty}) = 1 - p(x_t = 0 | \mathbf{x}_t^{-\infty})$. Here, the rate $\tilde{\lambda}(t)$ is computed as $\lambda(t)$ in Eq (69), but only with respect to past spikes that are emitted at times $\hat{t} < t$. This is because no spike that occurs within $[t, t + \Delta t)$ must be considered when computing $p_{\text{sim}}(x_t = 0 | \mathbf{x}_t^{-\infty})$.

For sufficiently long simulations, one can make use of the SLLN to compute the conditional entropy

$$H_{\text{sim}}(X|\mathbf{X}^{-\infty}) = -\frac{1}{N} \sum_{n=1}^{N} \log_2 p_{\text{sim}}(x_{t_n} | \mathbf{x}_{t_n}^{-\infty}), \qquad (74)$$

**Table 1. Simulation parameters of the GLIF-$\xi$ model.**

| Term | Description | Value | Units |
|---|---|---|---|
| $\lambda_0$ | Latency | 2.0 | $\text{ms}^{-1}$ |
| $1/\gamma$ | Membrane timescale | 15.3 | ms |
| $V_\infty$ | Equilibrium potential | -45.9 | mV |
| $V_0$ | Reset potential | -38.8 | mV |
| $V_T^*$ | Firing threshold baseline | -51.9 | mV |
| $\Delta V$ | Firing threshold sharpness | 0.75 | mV |
| $\alpha_\xi$ | Magnitude of the effective adaptation filter $\xi$ | 19.3 | mV |
| $\beta_\xi$ | Scaling exponent of the effective adaptation filter $\xi$ | 0.93 | - |
| $T_\xi$ | Cutoff of the effective adaptation filter $\xi$ | 8.3 | ms |
| $\delta t$ | Simulation step | 0.5 | ms |

The parameters were originally extracted from experimental recordings of (n = 14) L5 pyramidal neurons [43].

and thus the total history dependence

$$R_{\text{tot}} = 1 - \frac{H_{\text{sim}}(X|\boldsymbol{X}^{-\infty})}{\hat{H}(X)}, \tag{75}$$

which gives an upper bound to the history dependence for any past embedding.

**Computation of history dependence for given past embedding.** To compute history dependence for given past embedding, we use that the model neuron can be well approximated by a generalized linear model (GLM) within the parameter regime of our simulation. We can thus fit a GLM to the simulated data for the given past embedding $T, d, \kappa$ to obtain a good approximation of the corresponding true history dependence $R(T, d, \kappa)$. Note that this is a specific property if this model and does not hold in general. For example in experiments, we found that the GLM accounted for less history dependence than model-free estimates ([Fig 6]).

To map the model neuron to a GLM, we plug the membrane and threshold dynamics of Eqs ([67]) and ([70]) into the equation for the firing rate [Eq (69)], i.e.

$$\lambda(t) = \exp\left( \log \lambda_0 + V_\infty - V_T^* + \sum_{\hat{t}_j < t} \xi(t - \hat{t}_j) + (V_0 - V_\infty)e^{-\gamma(t - \hat{t}_0)} \right). \tag{76}$$

For the parameters used in the simulation, the decay time of the reset term $V_0 - V_\infty$ is $1/\gamma = 15.3$ ms. When compared to the minimum and mean inter-spike intervals of $\text{ISI}_{\min} = 25$ ms and $\overline{\text{ISI}} = 248$ ms, it is apparent that the probability for two spikes to occur within the decay time window is negligibly small. Therefore, one can safely approximate

$$(V_0 - V_\infty)e^{-\gamma(t - \hat{t}_0)} \approx \sum_{\hat{t}_j < t}(V_0 - V_\infty)e^{-\gamma(t - \hat{t}_j)}, \tag{77}$$

i.e. describing the potential reset after a spike as independent of other past spikes, because contributions beyond the last spike ($j > 0$) are effectively zero. Using the above approximation, one can formulate the rate as in a generalized linear model with

$$\lambda(t) = \exp\left( \mu + \sum_{j=1}^{d} h_j x_{t,j}^- \right), \tag{78}$$

where

$$\mu = \log \lambda_0 + V_\infty - V_T^* \tag{79}$$

$$h_j = \xi(j\delta t) + (V_0 - V_\infty)e^{-\gamma j \delta t}, \tag{80}$$

and $x_{t,j}^- \in \{0, 1\}$ indicates whether the neuron spiked in $[t - j\delta t, t - (j-1)\delta t]$. Therefore, the true spiking probability of the model is well described by a GLM.

We use this relation to approximate the history dependence $R(T, d, \kappa)$ for any past embedding $T, d, \kappa$ with a GLM with the same past embedding. Since in that case the parameters $\mu$ and $\boldsymbol{h}$ are not known, we fitted them to the simulated 900 minute recording via maximum likelihood (see above) and computed the history dependence according to

$$\hat{R}_{\text{GLM}}(T, d, \kappa) = 1 - \frac{\hat{H}_{\text{GLM}}(X|\boldsymbol{X}_{d,\kappa}^{-T})}{\hat{H}(X)}. \tag{81}$$

**Computation of history dependence as a function of the past range.** To approximate the model's true history dependence $R(T)$, for each $T$ we computed GLM estimates $\hat{R}_{\mathrm{GLM}}(T, d, \kappa)$ (Eq 81) for a varying number of past bins $d \in [25, 50, 75, 100, 125, 150]$. For each $d$, the scaling $\kappa$ was chosen such that the size of the first past bin was equal or less than 0.5 ms. To save computation time, and to reduce the effect of overfitting, the GLM parameters where fitted on 300 minutes of the simulation, whereas estimates $\hat{R}_{\mathrm{GLM}}(T, d, \kappa)$ were computed on the full 900 minutes of the simulated recording. For each $T$, we then chose the highest estimate $\hat{R}_{\mathrm{GLM}}(T, d, \kappa)$ among the estimates for different $d$ as the best estimate of the true $R(T)$.

## Experimental recordings

We analyzed neural spike trains from *in vitro* recordings of rat cortical cultures and salamander retina, as well as *in vivo* recordings in rat dorsal hippocampus (layer CA1) and mouse primary visual cortex. For all recordings, we only analyzed sorted units with firing rates between 0.5 Hz and 10 Hz to exclude the extremes of either inactive units or units with very high firing rate.

**Rat cortical culture.** Neurons were extracted from rat cortex (1 st day postpartum) and recorded *in vitro* on an electrode array 2–3 weeks after plating day. We took data from five consecutive sessions (`L_Prg035_txt_nounstim.txt`, `L_Prg036_txt_nounstim.txt`, ..., `L_Prg039_txt_nounstim.txt`) with a total duration of about $T_{\mathrm{rec}} \approx 203$ min. However, we only analyzed the first 90 minutes to make the results comparable to the other neural systems. We analyzed in total $n = 48$ sorted units that satisfied our requirement on the firing rate. More details on the recording procedure can be found in [67], and details on the data set proper can be found in [49].

**Salamander retina.** Spikes from larval tiger salamander retinal ganglion cells were recorded *in vitro* by extracting the entire retina on an electrode array [68], while a non-repeated natural movie (leaves moving in the wind) was projected onto the retina. The recording had a total length of about $T_{\mathrm{rec}} \approx 82$ min, and we analyzed in total $n = 111$ sorted units that satisfied our requirement on the firing rate. More details on the recording procedure and the data set can be found in [47, 48]. The spike recording was obtained from the Dryad database [47].

**Rat dorsal hippocampus (layer CA1).** We evaluated spike trains from a multichannel simultaneous recording made from layer CA1 of the right dorsal hippocampus of a Long-Evans rat during an open field task (data set ec014.277). The data set provided sorted spikes from 8 shanks with 64 channels. The recording had a total length of about $T_{\mathrm{rec}} \approx 90$ min. We analyzed in total $n = 28$ sorted units that were indicated as single units and satisfied our requirement on the firing rate. More details on the experimental procedure and the data set can be found in [45, 46]. The spike recording was obtained from the NSF-founded CRCNS data sharing website.

**Mouse primary visual cortex.** Neurons were recorded *in vivo* during spontaneous behavior, while face expressions were monitored. Recordings were obtained by 8 simultaneously implanted Neuropixel probes, and sorted units were located using the location of the electrode contacts provided in [50], and the Allen Mouse Common Coordinate Framework [69]. We analyzed in total $n = 142$ sorted units from the mouse "Waksman" that belonged to primary visual cortex (irrespective of their layer) and satisfied our requirement on the firing rate. Second, we only selected units that were recorded for more than $T_{\mathrm{rec}} \approx 40$ min (difference between the last and first recorded spike time). Details on the recording procedure and the data set can be found in [58] and [50].

## Parameters used for embedding optimization

The embedding dimension or number of bins was varied in a range $d \in [1, d_{\max}]$, where $d_{\max}$ was either $d_{\max} = 20$, $d_{\max} = 5$ (max five bins) or $d_{\max} = 1$ (one bin). During embedding optimization, we explored $N_\kappa = 10$ linearly spaced values of the exponential scaling $\kappa$ within a range $[0, \kappa_{\max}(d)]$. The maximum $\kappa_{\max}(d)$ was chosen for each number of bins $d \in [1, d_{\max}]$ such that the bin size of the first past bin was equal to a minimum bin size, i.e. $\tau_1 = \tau_{1,\min}$, which we chose to be equal to the time step $\tau_{1,\min} = \Delta t = 5$ ms. To save computation time, we did not consider any embeddings with $\kappa > 0$ if the past range $T$ and $d$ were such that $\tau_1(\kappa_{\max}(d)) \leq \Delta t$ for $\kappa = 0$. Similarly, for given $T$ and each $d$, we neglected values of $\kappa$ during embedding optimization if the difference $\Delta\kappa$ to the previous value of $\kappa$ was less than $\Delta\kappa_{\min} = 0.01$. In Table 2 we summarize the relevant parameters that were used for embedding optimization.

**Details to Fig 3.** For Fig 3B, the process was considered for $l = 1$ and an reactivation probability of $m = 0.8$. For $l = 1$, all probabilities can easily be calculated, with marginal probability to be active $p(x_t = 1) = h/(1 - m + mh)$, and conditional probabilities $p(x_t = 1|x_{t-1} = 1) = h + (1 - h)m$ and $p(x_t = 1|x_{t-1} = 0) = h$. From these probabilities, the total mutual information $I_{\mathrm{tot}}$ and total history dependence $R_{\mathrm{tot}}$ could be directly computed. We then plotted these quantities as a function of $h$, where values of $h$ were chosen to vary the firing rate between 0.5 and 10 Hz, with a bin size of $\Delta t = 5$ ms. For Fig 3C, the binary autoregressive process was simulated for $n = 10^7$ time steps with $m = 0.8$ ($l = 1$), whereas for $l = 5$, $m$ was adapted to yield approximately the same $R_{\mathrm{tot}}$ as for $l = 1$. The input activation probability $h$ was chosen to lead to a fixed probability $p(x = 1) \approx 0.025$, corresponding to 5 Hz firing rate with $\Delta t = 5$ ms. Autocorrelation $C(T)$ was computed using the MR.estimator toolbox [52], and $\Delta R(T)$ and $L(T)$ were estimated using plugin estimation. For Fig 3D, the same procedures were applied as in Fig 3C, but now $m$ was varied between 0.5 and 0.95, and $h$ was adapted for each $m$ to hold the firing rate fixed at 5 Hz. For Fig 3E, the same procedures were applied as in Fig 3C, but now $l$ was varied between 1 and 10, and $h$ and $m$ were adapted for each $l$ to hold the firing rate fixed at 5 Hz and $R_{\mathrm{tot}}$ fixed at the value for $l = 1$ and $m = 0.8$.

**Details to Fig 4A and 4B.** The branching process was simulated using the MR.estimator toolbox, with a time step of $\Delta t = 4$ ms, population rate of 500 Hz and subsampling probability of 0.01. Thus, the subsampled spike train had a firing rate of $\approx 5$ Hz. The branching parameter was set to $m = 0.98$ with analytic autocorrelation time $\tau_C(m) = 198$ ms. For a long simulation, autocorrelation $C(T)$ was computed using the MR.estimator toolbox, $L(T)$ using plugin

**Table 2. Parameters used for embedding optimization.**

| Symbol | Value | Settings variable name | Description |
|---|---|---|---|
| $\Delta t$ | 0.005 | `embedding_step_size` | Time step (in seconds) for the discretization of neural spiking activity. |
| $d$ | $1, 2, \ldots, d_{\max}$ | `embedding_number_of_bins_set` | Set of embedding dimensions. |
| $N_\kappa$ | 10 | `number_of_scalings` | Number of linearly spaced values of the exponential scaling $\kappa$. |
| $\tau_{1,\min}$ | 0.005 | `min_first_bin_size` | Minimum bin size (in seconds) of the first past bin. |
| $\Delta\kappa_{\min}$ | 0.01 | `min_step_for_scaling` | Minimum required difference between two values of $\kappa$. |
| $p$ | 0.05 | `bbc_tolerance` | Tolerance for the acceptance of estimates for BBC. |
| - | False | `cross_validated_optimization` | Is cross-validation used for optimization or not. |
| - | 250 | `number_of_bootstraps_R_max` | Number of bootstrap samples used to estimate $\sigma_{\hat{R}_{\max}}$. |
| $l$ | $1/r\Delta t$ | `block_length_l` | Block length used for blocks-of-blocks bootstrapping. |
| - | all | `estimation_method` | Estimators for which embeddings are optimized (BBC, Shuffling) |

To facilitate reproduction, we added the settings variable names of the parameters as they are used in the toolbox [37].

estimation, and $R(T)$ using the embedding-optimized Shuffling estimator with $d_{max} = 20$. The generalized timescales $\tau_R$ and $\tau_L$ were computed with $T_0 = 10$ ms.

**Details to Fig 4C and 4D.** The Izhikevich model was simulated with the PyNN toolbox [70], with parameters set to the chattering mode ($a = 0.02$, $b = 0.2$, $c = -50$, $d = 2$), simulation time bin $dt = 0.01$ ms, and noisy input with mean 0.011 and standard deviation 0.001. For the analysis, a time step of $\Delta t = 1$ ms was chosen. Apart from that, $C(T)$ and $L(T)$ were computed as for Fig 4B. Here, $R(T)$ was computed with BBC and $d_{max} = 20$, which revealed higher $R_{tot}$ than Shuffling. To compute $\tau_R$, we set $T_0 = 0$.

**Details to Fig 4E and 4F.** The GLIF model was simulated as described in Benchmark neuron model (model with 22 s past kernel). The analysis time step was $\Delta t = 5$ ms. Apart from that, $C(T)$ and $L(T)$ were computed as for Fig 4B. History dependence $R(T)$ was estimated using a GLM as described in Benchmark neuron model. To compute $\tau_R$, we set $T_0 = 10$ ms.

**Details to Fig 5A and 5B.** In Fig 5A and 5B, we applied the ML, NSB, BBC and Shuffling estimators of $R(\tau, d)$ to a simulated recording of 90 minutes. Embedding parameters were $T = d \cdot \tau$ and $\kappa = 0$, with $\tau = 20$ ms and $d \in [1, 60]$. Since the goal was to show the properties of the estimators, confidence intervals were estimated from 50 repeated 90 minute simulations instead of bootstrap samples from the same recording. Each simulation had a burning in period of 100 seconds. To estimate the true $R(\tau, d)$, a GLM was fitted on a 300 minute recording and evaluated on the full 900 minute recording for the estimation of $R$.

**Details to Fig 5C.** In Fig 5C, history dependence $R(T)$ was estimated on a 90 minute recording for 57 different values of $T$ in a range $T \in [10$ ms, 3 s]. Embedding-optimized estimates were computed with up to $d_{max} = 25$ past bins, and 95% confidence intervals were computed using the standard deviation over $n = 100$ bootstrap samples (see Bootstrap confidence intervals). To estimate the true $R(T, d^*, \kappa^*)$ for the optimized embedding parameters $d^*$, $\kappa^*$ with either BBC or Shuffling, a GLM was fitted for the same embedding parameters on a 300 minute recording and evaluated on 900 minutes recording for the estimation of $R$.

**Details to Fig 6.** For Fig 6, history dependence $R(T)$ was estimated for 61 different values of $T$ in a range $T \in [10$ ms, 5 s]. For each recording, we only analyzed the first 90 minutes to have a comparable recording length. For embedding optimization, we used $d_{max} = 20$ as a default for BBC and Shuffling, and compared the estimates with the Shuffling estimator optimized for $d_{max} = 5$ (max five bins) and $d_{max} = 1$ (one bin). For the GLM, we only estimated $R(T_D)$ for the temporal depth $T_D$ that was estimated with BBC. To optimize the estimate, we computed GLM estimates of $R(T_D)$ with the optimal embedding found by BBC, and for varying embedding dimension $d \in [1, 2, 3, .., 20, 25, 30, 35, 40, 45, 50]$, where for each $d$ we chose $\kappa$ such that $\tau_1 = \Delta t$. We then chose the embedding that minimized the BIC, and took the corresponding estimate $\hat{R}(T_D)$ as a best estimate for $R_{tot}$. For Fig 6A, we plotted only spike trains of channels that were identified as single units. For Fig 6B, 95% confidence intervals were computed using the standard deviation over $n = 100$ bootstrap samples. For Fig 6C, embedding-optimized estimates with uniform embedding ($\kappa = 0$) were computed with $d_{max} = 20$ (BBC and Shuffling) or $d_{max} = 5$ (Shuffling). Medians were computed over the $n = 28$ sorted units in CA1.

**Details to Figs 7 and 8.** For Figs 7 and 8, history dependence was $R(T)$ was estimated for 61 different values of $T$ in a range $T \in [10$ ms, 5 s] using the Shuffling estimator with $d_{max} = 5$. The autocorrelation coefficients $C(T)$ were computed with the MR.Estimator toolbox [52], and the autocorrelation time $\tau_C$ was obtained using the `exponential_offset` fitting function. For each recording, we only analyzed the first 40 minutes to have a comparable recording length. For Fig 7, medians of $\tau_R$, $\tau_C$ and $R_{tot}$ were computed over all sorted units that were analyzed, and 95% confidence intervals on the medians were obtained by bootstrapping

with $n = 10000$ resamples of the median. For Fig 8, 95% confidence intervals were computed using the standard deviation over $n = 100$ bootstrap samples.

## Practical guidelines: How to estimate history dependence from neural spike recordings

Estimating history dependence (or any complex statistical dependency) for neural data is notoriously difficult. In the following, we address the main requirements for a practical and meaningful analysis of history dependence, and provide guidelines on how to fulfill these requirements using embedding optimization. A toolbox for Python3 is available online [37], together with default parameters that worked best with respect to the following requirements. It is important that practitioners make sure that their data fulfill the data requirements (points 4 and 5).

**1) The embedding of past spiking activity should be individually optimized to account for very different spiking statistics.** It is crucial to optimize the embedding for each neuron individually, because history dependence can strongly differ for neurons from different areas or neural systems (Fig 7), or even among neurons within a single area (see examples in Fig 8). Individual optimization enables a meaningful comparison of information timescale and history dependency $R$ between neurons.

**2) The estimation has to capture any non-linear or higher-order statistical dependencies.** Embedding optimization using both, the BBC or Shuffling estimators, is based on non-parametric estimation, in which the joint probabilities of current and past spiking are directly estimated from data. Thereby, it can account for any higher-order or non-linear dependency among all bins. In contrast, the classical generalized linear model (GLM) that is commonly used to model statistical dependencies in neural spiking activity [20, 21] does not account for higher-order dependencies. We found that the GLM recovered consistently less total history dependence $R_{tot}$ (Fig 6D). Hence, to capture single-neuron history dependence, higher-order and non-linear dependencies are important, and thus a non-parametric approach is advantageous.

**3) Estimation has to be computationally feasible even for a high number of recorded neurons.** Strikingly, while higher-order and non-linear dependencies are important, the estimation of history dependence does not require high temporal resolution. Optimizing up to $d_{max} = 5$ past bins with variable exponential scaling $\kappa$ could account for most of the total history dependence that was estimated with up to $d_{max} = 20$ bins (Fig 6D). With this reduced setup, embedding optimization is feasible within reasonable computation time. Computing embedding-optimized estimates of the history dependence $R(T)$ for 61 different values of $T$ (for 40 minute recordings, the approach used for Figs 7 and 8) took around 10 minutes for the Shuffling estimator, and about 8.5 minutes for the BBC per neuron on a single computing node. Therefore, we recommend using $d_{max} = 5$ past bins when computation time is a constraint. Ideally, however, one should check for a few recordings if higher choices of $d_{max}$ lead to different results, in order to cross-validate the choice of $d_{max} = 5$ for the given data set.

**4) Estimates have to be reliable lower bounds, otherwise one cannot interpret the results.** It is required that embedding-optimized estimates do not systematically overestimate history dependence for any given embedding. Otherwise, one cannot guarantee that *on average* estimates are lower bounds to the total history dependence, and that an increase in history dependence for higher past ranges is not simply caused by overestimation. This guarantee is an important aspect for the interpretation of the results.

For BBC, we found that embedding-optimized estimates are unbiased if the variance of estimators is sufficiently small (S1 Fig). The variance was sufficiently small for recordings of 90

minutes duration. When the variance was too high (short recordings with 3–45 minutes recording length), maximizing estimates for different embedding parameters introduced very mild overestimation due to overfitting (1–3%) (S1 Fig). The overfitting can, however, be avoided by cross-validation, i.e. optimizing the embedding on one half of the recording and computing estimates on the other half. *Using cross-validation*, we found that embedding-optimized BBC estimates were unbiased even for recordings as short as 3 minutes (S1 Fig).

For Shuffling, we also observed overfitting, but the overestimation was small compared to the inherent systematic underestimation of Shuffling estimates. Therefore, we observed no systematic overestimation by embedding-optimized Shuffling estimates on the model neuron, even for shorter recordings (3 minutes and more). Thus, for the Shuffling estimator, we advice to apply the estimator without cross-validation as long as recordings are sufficiently long (10 minutes and more, see next point).

**5) Spike recordings must be sufficiently long (at least 10 minutes), and of similar length, in order to allow for a meaningful comparison of total history dependence and information timescale across experiments.** The recording length affects estimates of the total history dependence $R_{\text{tot}}$, and especially of the information timescale $\tau_R$. This is because more data allow more-complex embeddings, hence higher history dependence can be estimated. Moreover, complex embeddings are particular relevant for long past ranges $T$. Therefore, if recordings are shorter, smaller $R(T)$ will be estimated for long past ranges $T$, leading to smaller estimates of $\tau_R$. We found that for shorter recordings, estimates of $R_{\text{tot}}$ were roughly the same as for 90 minutes, but estimates of $\tau_R$ were considerably smaller (S2 and S3 Figs).

To enable a meaningful comparison of the information timescale between neurons, one thus has to ensure that recordings are sufficiently long (in our experience at least 10 minutes), otherwise differences in $\tau_R$ may not be well resolved. Below 10 minutes, we found that estimates of $\tau_R$ could be less than half of the value that was estimated for 90 minutes, and also estimates of $R_{\text{tot}}$ showed a notable decrease. In addition, all recordings should have comparable length to prevent that differences in history dependence or timescale are due to different recording lengths.

## Supporting information

**S1 Fig. Embedding optimization leads to mild overfitting for short recordings, which can be avoided by cross-validation.** Shown is the relative bias for two versions of the GLIF model with spike adaption, one with 1 s and the other with 22 s past kernel. The relative bias refers to the relative difference between embedding-optimized estimates $\hat{R}(T, d^*, \kappa^*)$ and the model's true history dependence $R(T, d^*, \kappa^*)$ for the same optimized embedding parameters $d^*, \kappa^*$. The relative bias for $\hat{R}_{\text{tot}}$ was computed by first averaging the relative difference $(\hat{R}(T, d^*, \kappa^*) - R(T, d^*, \kappa^*))/R(T, d^*, \kappa^*)$ for $T \in [T_D, T_{max}]$, and second averaging again over 30 different simulations for $T_{\text{rec}}$ between 1 and 20 minutes, and 10 different simulations for 45 and 90 minutes. Embedding parameters were optimized for each simulation, respectively, using parameters as in Table 2 with $d_{\max} = 25$. (Left) For BBC, the relative bias for $\hat{R}_{\text{tot}}$ is zero only if recordings are sufficiently long ($> 20$ minutes for 1 s kernel, and $\approx 90$ minutes for 22 s kernel). When recordings are shorter, the relative bias increases, and thus estimates are mildly overestimating the model's true history dependence for the optimized embedding parameters. For Shuffling, estimates provide lower bounds to the model's true history dependence, hence the relative bias remains negative even in the presence of overfitting. (Right) When one round of cross-validation is applied, i.e. embedding parameters are optimized on the first, and estimates are computed on the second half of the data, the bias is approximately zero for BBC even for short recordings, or more negative for the Shuffling estimator. Therefore, we conclude that the

origin of overfitting is the selection of embedding parameters on the same data that are used for the estimation of $R$. Errorbars show 95% bootstrap confidence intervals on the mean over $n = 10$ (45 or 90 min) or $n = 30$ ($\leq$ 20 min) different simulations.
(TIF)

**S2 Fig. For the simulated neuron model, recording length has little effect on the estimated total history dependence, but large impact on the estimated information timescale.** (Left) Mean estimated total history dependence $\hat{R}_{\text{tot}}$ for different recording lengths, relative to the true total history dependence $R_{\text{tot}}$ of the model (GLIF with spike adaption with 1 s or 22 s past kernel). As the recording length decreases, so does $\hat{R}_{\text{tot}}$. However, with only 3 minutes, one does still infer about $\approx$ 95% of the true $R_{\text{tot}}$. (Right) In contrast, the estimated information timescale $\hat{\tau}_R$ decreases strongly with decreasing recording length. With 3 minutes and less, only $\approx$ 75% of the true $\tau_R$ is estimated on average. Note that for the simpler 1 s model (top), an accurate estimation of the true $\tau_R$ is possible for 90 minute recordings, whereas for the 22 s model (bottom), the estimated $\hat{\tau}_R$ remains below the true value. Shown are mean values for 30 different simulations for $T_{\text{rec}}$ between 1 and 20 minutes, and 10 different simulations for 45 and 90 minutes, as well as 95% confidence intervals on the mean based on bootstrapping.
(TIF)

**S3 Fig. For experimental data, too, recording length has little effect on estimated total history dependence, but larger impact on the estimated information timescale.** (Left) Total history dependence $R_{\text{tot}}$ for different recording lengths, relative to the total history dependence estimated for a 90 minute recording. As long as recordings are 10 minutes or longer, one does still estimate about $\approx$ 95% as much or more of $R_{\text{tot}}$ as for 90 minutes, for all three recordings. For less than 10 minutes, the estimated total history dependence decreases down to 90% (CA1), or increases again due to overfitting (retina). (Right) Similar to the GLIF model, the estimated information timescale $\tau_R$ decreases more strongly with decreasing recording length. With 10 minutes and more, one estimates around $\approx$ 75% or more of the $\tau_R$ that is estimated on a 90 minute recording. Note that for the experimental data, the estimated timescale of the BBC estimator depends more strongly on the recording time, whereas the Shuffling estimator is more robust, especially for $d_{\text{max}} = 5$. Shown is the median with 95% bootstrap confidence intervals over $n = 10$ randomly chosen sorted units for each neural system. Before taking the median over sorted units, for each unit we averaged estimates over 10 excerpts of the full recording, each with 3 or 5 minutes duration, and over 8,4 and 2 excerpts with 10, 20 and 45 minutes duration, respectively.
(TIF)

**S4 Fig. Example estimation results for the generalized leaky integrate-and-fire model (GLIF) with 1 s past kernel.** For each recording length, we show the embedding-optimized estimates of history dependence $R(T)$ with and without cross-validation, for BBC (red) and Shuffling (blue) with $d_{\text{max}} = 25$, as well as the ground truth for the same embeddings that were found during optimization (dashed lines). Dashed lines indicate the estimated information timescale $\hat{\tau}_R$ and total history dependence $\hat{R}_{\text{tot}}$. Shaded areas indicate $\pm$ two standard deviations obtained by bootstrapping.
(TIF)

**S5 Fig. Example estimation results for the generalized leaky integrate-and-fire model (GLIF) with 22 s past kernel.** For each recording length, we show the embedding-optimized estimates of history dependence $R(T)$ with and without cross-validation, for BBC (red) and Shuffling (blue) with $d_{\text{max}} = 25$, as well as the ground truth for the same embeddings that were

found during optimization (dashed lines). Dashed lines indicate the estimated information timescale $\hat{\tau}_R$ and total history dependence $\hat{R}_{\text{tot}}$. Shaded areas indicate ± two standard deviations obtained by bootstrapping.
(TIF)

**S6 Fig. Estimation results for all sorted units in rat dorsal hippocampus (layer CA1).** For each unit, we show the embedding-optimized estimates of history dependence $R(T)$ for BBC with $d_{\max} = 20$ (red), as well as Shuffling with $d_{\max} = 20$ (blue), $d_{\max} = 5$ (green) and $d_{\max} = 1$ (yellow). Dashed lines indicate estimates of the information timescale $\tau_R$ and total history dependence $R_{\text{tot}}$. Also shown is the embedding-optimized GLM estimate (violet square) with a past range equal to the temporal depth that was found with the BBC estimator.
(TIF)

**S7 Fig. Estimation results for all sorted units in rat cortical culture.** For each unit, we show the embedding-optimized estimates of history dependence $R(T)$ for BBC with $d_{\max} = 20$ (red), as well as Shuffling with $d_{\max} = 20$ (blue), $d_{\max} = 5$ (green) and $d_{\max} = 1$ (yellow). Dashed lines indicate estimates of the information timescale $\tau_R$ and total history dependence $R_{\text{tot}}$. Also shown is the embedding-optimized GLM estimate (violet square) with a past range equal to the temporal depth that was found with the BBC estimator.
(TIF)

**S8 Fig. Estimation results for all sorted units in salamander retina.** For each unit, we show the embedding-optimized estimates of history dependence $R(T)$ for BBC with $d_{\max} = 20$ (red), as well as Shuffling with $d_{\max} = 20$ (blue), $d_{\max} = 5$ (green) and $d_{\max} = 1$ (yellow). Dashed lines indicate estimates of the information timescale $\tau_R$ and total history dependence $R_{\text{tot}}$. Also shown is the embedding-optimized GLM estimate (violet square) with a past range equal to the temporal depth that was found with the BBC estimator.
(TIF)

**S9 Fig. Estimation results for all sorted units in mouse primary visual cortex.** For each unit, we show the embedding-optimized Shuffling estimates of history dependence $R(T)$ for $d_{\max} = 5$. Dashed lines indicate estimates of the information timescale $\tau_R$ and total history dependence $R_{\text{tot}}$.
(TIF)

**S10 Fig. Bootstrapping yields accurate estimates of standard deviation and confidence intervals.** (Left) Shown is the standard deviation on BBC estimates (blue) obtained from 250 "blocks of blocks" bootstrap samples on a single recording (GLIF model with 22 s past kernel). It agrees well with the true standard deviation (black), which we estimated from 100 repeated simulations of the same recording length and embedding. As expected, the standard deviation decreases substantially for longer recordings. For each recording length, estimates were computed for typical optimal embedding parameters $d^*$, $\kappa^*$ and $T = T_D$ that were found by embedding optimization. Errorbars show mean and standard deviation of the estimated $\sigma(R)$ over the repeated simulations. (Right) The 95% confidence intervals based on two standard deviations $\sigma(R)$ have approximately the claimed confidence level (CI accuracy). Standard deviation was estimated from 250 "blocks of blocks" bootstrap samples. For each recording length, we computed estimates $\hat{R}$ and the bootstrap confidence intervals on the 100 simulations. We then computed the confidence level (CI accuracy) by counting how often the true value of $R$ was contained in the estimated confidence interval (green line). Estimates and the true value of $R$ were computed for the same typical embedding parameters $d^*$, $\kappa^*$ and $T = T_D$ as before.
(TIF)

**S11 Fig. Total history dependence and information timescale for increasing branching parameter $m$.** Similar to the binary autoregressive process, increasing the branching parameter $m$ increases the total history dependence $R_{tot}$, whereas the information timescale $\tau_R$ stays constant, or even decreases for high $m$. For each $m$, the input activation probability $h$ was adapted to hold the firing rate fixed at 5 Hz.
(TIF)

**S12 Fig. The estimated information timescale varies between estimators.** For each sorted unit (grey dots), estimates of the information timescale $\tau_R$ are plotted relative to the corresponding BBC estimate for $d_{max} = 20$. The BBC estimator tends to estimate higher timescales than the Shuffling estimator on recordings of CA1 and cortical culture, whereas for retina the medians of different estimators are more similar. Although estimates of the timescale are highly variable between estimators, Shuffling with only $d_{max} = 5$ past bins still estimates timescales of at least 80% of the timescales that are estimated with BBC. Errorbars indicate median over sorted units and 95% bootstrap confidence intervals on the median.
(TIF)

**S13 Fig. Total history dependence and information timescale show no clear dependence on the firing rate, whereas the total mutual information tends to increase with the rate.** Shown are the same estimates of the total history dependence $R_{tot}$ and information timescale $\tau_R$ as in [Fig 7](Shuffling estimator with $d_{max} = 5$) versus the firing rates of sorted units (dots). The total mutual information $I_{tot}$ is equal to $R_{tot}$ times the spiking entropy $H$(spiking) of the respective unit. While $I_{tot}$ tends to increase with firing rate, no clear relation is visible for $R_{tot}$ or $\tau_R$. Errorbars indicate median over sorted units and 95% bootstrap confidence intervals on the median.
(TIF)

**S14 Fig. Relationship between total history dependence or information timescale and standard statistical measures of neural spike trains.** Estimates of the total history dependence $R_{tot}$ tend to decrease with the median interspike interval (ISI), and to increase with the coefficient of variation $C_V$. This result is expected for a measure of history dependence, because a shorter median ISI indicates that spikes tend to occur together, and a higher $C_V$ indicates a deviation from independent Poisson spiking. In contrast, the information timescale $\tau_R$ tends to increase with the autocorrelation time, as expected, with no clear relation to the median ISI or the coefficient of variation $C_V$. However, the correlation between the measures depends on the neural system. For example in retina ($n = 111$), $R_{tot}$ is significantly anti-correlated with the median ISI (Pearson correlation coefficient: $r = -0.69$, $p < 10^{-5}$) and strongly correlated with the coefficient of variation $C_V$ ($r = 0.90$, $p < 10^{-5}$), and $\tau_R$ is significantly correlated with the autocorrelation time $\tau_C$ ($r = 0.75$, $p < 10^{-5}$). In contrast, for mouse primary visual cortex ($n = 142$), we found no significant correlations between any of these measures. Results are shown for the Shuffling estimator with $d_{max} = 5$, and $T_0 = 10$ ms. Errorbars indicate median over sorted units and 95% bootstrap confidence intervals on the median.
(TIF)

**S15 Fig. Excluding short-term contributions helps to differentiate the timescales for different neural systems.** By only considering gains $\Delta R(T)$ for past ranges $T > T_0$ when computing the information timescale $\tau_R$, short-term effects that are related to the refractory period and different firing modes are excluded. The higher $T_0$, the higher is the distance in the median $\tau_R$ between systems (especially between salamander retina and mouse primary visual cortex). This is because both timescales $\tau_R$ and $\tau_C$ increase with $T_0$ for CA1 and primary visual cortex, whereas they decrease for retina. The same holds for the autocorrelation time $\tau_C$, where only

time lags $T > T_0$ were considered when fitting an exponential decay to the autocorrelograms. Note that if the decay is perfectly exponential, then $T_0$ does not affect the results. Estimates of $R_{tot}$ and $\tau_R$ are shown for the Shuffling estimator with $d_{max} = 5$. Errorbars indicate median over sorted units and 95% bootstrap confidence intervals on the median.
(TIF)

**S16 Fig. Total history dependence decreases for small time bins $\Delta t$.** The choice of the time bin $\Delta t$ of the spiking activity has little effect on the information timescale $\tau_R$, whereas the total history dependence $R_{tot}$ decreases for small time bins $\Delta t < 5$ ms. This is consistent across experiments. The smaller the time bin, the higher the risk that noise in the spike emission reduces the overall predictability or history dependence in the spiking, whereas an overly large time bin holds the risk of destroying coding relevant time information in the spike train. Thus, we chose the smallest time bin $\Delta t = 5$ ms that does not yet show a substantial decrease in $R_{tot}$. We do not plot results for higher $\Delta t$, because for higher $\Delta t$ we observed many instances of multiple spikes in the same time bin. Results are shown for the Shuffling estimator with $d_{max} = 5$, and $T_0 = 10$ ms. Errorbars indicate median over sorted units and 95% bootstrap confidence intervals on the median.
(TIF)

## Acknowledgments

We thank the Priesemann group, especially Matthias Loidolt, Fabian Mikulasch, Andreas Schneider, David A. Ehrlich, Jonas Dehning, F. Paul Spitzner, as well as Abdullah Makkeh for valuable comments and for reviewing the manuscript.

## Author Contributions

**Conceptualization:** Lucas Rudelt, Michael Wibral, Viola Priesemann.

**Formal analysis:** Lucas Rudelt.

**Funding acquisition:** Viola Priesemann.

**Investigation:** Lucas Rudelt.

**Methodology:** Lucas Rudelt, Daniel González Marx, Viola Priesemann.

**Resources:** Viola Priesemann.

**Software:** Lucas Rudelt, Daniel González Marx.

**Supervision:** Michael Wibral, Viola Priesemann.

**Visualization:** Lucas Rudelt, Daniel González Marx.

**Writing – original draft:** Lucas Rudelt.

**Writing – review & editing:** Lucas Rudelt, Daniel González Marx, Michael Wibral, Viola Priesemann.

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
