## [Decision Letter · Decision Letter 0]

1 Dec 2020

Dear Mr. Rudelt,

Thank you very much for submitting your manuscript "Embedding optimization reveals long-lasting history dependence in neural spiking activity" for consideration at PLOS Computational Biology.

As with all papers reviewed by the journal, your manuscript was reviewed by members of the editorial board and by several independent reviewers.

While the methods look overall correct, with some needed improvements, what is less clear is the potential for application and the impact. Please work on that (of course not overselling it but critically situating it in the current panorama). A resubmission addressing all the outstanding issues could be offered publication in PLOS Computational Biology if the scope is clearer, or in PLOS One if it is not. In this latter case you could of course decline the offer.

In light of the reviews (below this email), we would like to invite the resubmission of a significantly-revised version that takes into account the reviewers' comments.

We cannot make any decision about publication until we have seen the revised manuscript and your response to the reviewers' comments. Your revised manuscript is also likely to be sent to reviewers for further evaluation.

Sincerely,

Daniele Marinazzo

Deputy Editor

PLOS Computational Biology

Reviewer's Responses to Questions

**Comments to the Authors:**

Reviewer #1: This paper proposes new metrics for measuring history dependence in neural spike trains, and uses a particular coarse-graining in combination with existing entropy/mutual information estimation methods to estimate this metric for a range of neural spike trains. The authors then try to draw conclusions about their estimated metrics for various real neural spike trains.

The methods aspect of this seems relatively sound. I do have a suggestion for the authors, though, in terms of presentation: I'd put the vast majority of the methods in the Methods rather than the Results section. Basically, the discussion of the curse of dimensionality and the Data Processing Inequality in various forms (large number of bins is curse of dimensionality, can lead to overestimation and small number of bins yields lower MI due to Data Processing Inequality) seem to me to be well-worn statistical ground and not worthy of so much of the Results section. I'd also emphasize more that your main contribution to estimation of these information quantities is a particularly clever coarse-graining that assumes the recency hypothesis.

But that's not my main worry. I'm mainly worried that the metric isn't necessarily the right one for the job. On the chopping block is not just your R(T) (which I would not call a redundancy, but rather just a version of the predictive information divided by) and T_D (which I have a few comments on later), but also the autocorrelation function (which you discard, for reasons that make sense) and the predictive information (which you essentially have a version of in your numerator, but see Nemenman et al) and all the information measures in "Anatomy of a Bit" by Ryan James et al. Based on my experience playing with these metrics, I'd say the following:

-- it is likely that T_D will grow with the size of your data set, and so what's really relevant is the rate of growth; that may be a better way to distinguish between different time series;

-- it is likely that R(T) has some weird behavior with the time bin size for the present neural patterning that has not yet been discussed and should be;

-- I still have no idea how or if either R(T) or T_D(data set size) capture anything related to history dependence.

Before I recommend acceptance, I would ask for simulations of an Izhikevich neuron that can adopt different neuron types. The strawmen, in my opinion, should be first the autocorrelation function and then the predictive information. I believe that information measures of time series can reveal the type of neuron or aspects of how it behaves, but I don't see why I should switch from using the predictive information to using R(T) or its relative T_D. What am I getting from R(T) that I'm not getting from predictive information? What is the intuition behind introducing this new measure? What do the authors even mean by "history dependence"? If I am to normalize something like predictive information by single symbol entropy, as the authors do here, what neural spike train do I now correctly classify as having long history dependence that I before believed had little history dependence? As I am missing this intuition from the paper, I cannot recommend acceptance-- yet.

Smaller things:

-- I would not say that this measure of history dependence has anything to do with the efficient coding hypothesis, which is more about how stimulus is transformed by a neuron so that the neuron has maximal entropy, or sometimes (depending on who's using the term) is about how mutual information between stimulus and neuron is close to the entropy of the neural activity;

-- I would add some words on when your embedding method is likely to fail, which is precisely when initial conditions really really matter and the recency hypothesis is inaccurate-- e.g. network of Izhikevich neurons-- and which (notably) some might call long-term history dependence.

If the authors can convince me that their metric R(T) and its relative T_D (which should really be some aspect of how T_D changes with recording length) contain useful information that stumps the predictive information, then I will happily recommend acceptance.

Reviewer #2: This paper is a potentially important contribution to neuroscientific

toolbox. The authors propose an extension of existing information

theoretic approaches that allows for an unbiased estimation of a

neuron's history dependence on temporal depth and history dependence.

The paper presents a thorough approach to controlling bias and

overfitting. Further, the method is applied to several open datasets and

an intriguing finding is described. Finally, the code to apply the

methods described in the paper is made available with thorough

documentation.

I am enthusiastic but have one minor concern and a few related requests

for additional analyses described below. In addition, I made a pull

request on Github that may help improve the usability of this tool;

hopefully, the authors will build on it to include a few tests of the

code. This is not a requirement for this review, but it would be great

to see code coverage increase to >50%.

The concern is the following. History dependence R depends on the

entropy of current spiking conditional on the past, as well as on the

entropy of current spiking. The average firing rate of a neuron changes

its entropy; presumably, this is the reason that entropy of current

spiking is in the denominator. In theory, the product does not depend on

the neuron's average firing rate; however, it would be nice to get a

demonstration that R_tot or T_D do not vary as a function of the GLIF

neuron's average firing rate, median ISI, or CV. More importantly, I'd

like to see a scatterplot of these quantities vs R_tot and T_D in the

datasets from Fig. 5. If authors find no correlation there, it may be

instructive to look for a different connection to traditional statistics

as described in the first section of Discussion. Surely, we won't find

any perfect replacements for history dependence, but if T_D is loosely

related to some function of autocorrelation, it will help ground

researchers in more familiar terrain.

The following lists a few minor suggestions.

In most citations, the name of the journal is missing. Is this by

design?

In line 78, what does 'discrete past embedding of spiking activity'

mean? Do you refer to a 'reduced representation' of the past, or the

discrete nature of spiking data? I am trying to discern whether past

embedding with binary data has been described in practical terms before.

In line 152, you may wish to say something like 'while minimizing the

risk of overestimation'.

Line 163 mentions errorbars, but none are visible in Fig. 2D. I think a

different place in the paper mentions 2xSTD errorbars being too small to

be visible, but does that come later?

I am confused regarding the status of GLM in this paper. Line 337 justly

points out its systematic underestimation of history dependence, while

line 194 claims that the authors used GLM as ground truth for R(T, d,

kappa). Please clarify.

In Fig 4, why are bootstrapping errorbars not centered around the median

(bars' height)?

When refering to results from extracellular recordings, it may be best

to call the units identified through spikesorting "single units" rather

than "neurons" to remind us that spikesorting is somewhat subjective.

In Fig 5, would it be possible to include a scatterplot of history

dependence estimated from GLM?

Please attempt to interpret the results of Fig 6 further. Why is it that

single unit 3 has such a distinctive shape? What might this mean for the

corresponding neuron's information processing? What follow-up would you

suggest for researchers using your tool when they see shapes like these?

Would inspecting autocorrelograms help? Include any diagnostic

information you find helpful.

Related: What is the interpretation of a peak followed by decay in R(T)

as in Fig S7, row 2, middle two?

There is a typo in line 1298 and in caption to S4.

The sentence that starts on line 1326 is too long. Also, it may be good

to italicize 'blocks of blocks' here.

Reviewer #3: Embedding optimization reveals long-lasting history dependence in neural spiking

Activity

• Summary of the paper and novelties

This work investigates how to reliably quantify the dependence of a single neuron's spiking on its own preceding activity, called history dependence. Previous studies used limited representations of past activity (the so-called past embedding) to estimate information theory-based measures. Here it is argued that a careful embedding of past activity is crucial.

A novel embedding-optimization method is proposed here that optimizes temporal binning of past spiking to capture most of the magnitude and the temporal depth of history dependence. The new method is validated against simulated data of a LIF neuron model and empirical data from different databases that account for a large variety of spiking statistics.

• Strengths

The main strengths of the work are:

- It is demonstrated that previous ad hoc embedding strategies are likely to capture much less history dependence, or lead to estimates that severely overestimate the true history dependence. The new method maximizes the estimated history dependence while avoiding overestimation.

- The new method is flexible enough to account for the variety of spiking statistics encountered in experiments.

• Weaknesses and suggestions

A weak point of the work is that for spike trains with long temporal depths (e.g., larger than 3 seconds, as in Fig. 3 C), the temporal depth estimated by the optimization method is much smaller (630 ms). This is a critical point to discuss in terms of possible limitations to estimate the timescale of neural processing at different stages of the brain.

Another drawback of the new optimization methods is that they perform worse on short recordings: the estimated history dependence is overestimated when applying BBC to recordings of 3 minutes (S1 Fig) and the estimated temporal depth is underestimated to half of the real temporal depth (S2 Fig). This aspect might be discussed in the paper, analyzing possible limitations on application of optimization techniques to experimental data of short length.

Some minor suggestions:

- Line 60: Could you comment on why the time bin of current spiking is chosen to be 5 ms?

- Fig. 1 it is included in the Methods summary but is not well described in the text. Either move it to Methods, or further explain it here. In the figure caption, please provide more details of the figure, e.g., explain what is ML, NSB and BBC.

- Fig S1 and paragraph between lines 272 and 286: how is each half of the data selected for cross-validation? Are multiple rounds of cross-validation performed using different partitions (in this case different halves) of the data?

- Fig 4C: why BBC is computed with d = 20, and shuffling with d = 5?

- Fig S4 is not referenced in the text.

- Typo: “errorbars” instead of “error bars” (for example, in line 262).

- The publication year is missing in references.

Methods are written in an appropriate and informative way

The paper is well written and concepts are provided in a correct, clear and suitable way.

**Have all data underlying the figures and results presented in the manuscript been provided?**

Reviewer #1: Yes

Reviewer #2: Yes

Reviewer #3: Yes

PLOS authors have the option to publish the peer review history of their article (what does this mean?). If published, this will include your full peer review and any attached files.

Reviewer #1: No

Reviewer #2: No

Reviewer #3: No
---

## [Decision Letter · Decision Letter 1]

31 Mar 2021

Dear Mr. Rudelt,

We are pleased to inform you that your manuscript 'Embedding optimization reveals long-lasting history dependence in neural spiking activity' has been provisionally accepted for publication in PLOS Computational Biology.

Best regards,

Daniele Marinazzo

Deputy Editor

PLOS Computational Biology

Daniele Marinazzo

Deputy Editor

PLOS Computational Biology

Reviewer's Responses to Questions

**Comments to the Authors:**

Reviewer #1: The review is uploaded as an attachment-- a pdf with comments. My general comments are that this manuscript is now much improved. The examples have convinced me that R(T) might be of interest to the neuroscience community, and it is clear that the authors have done their due diligence in estimating it as well as possible. I'm less sold on \\tau_R, especially since it just seems to be correlated with \\tau_C, but I have no fundamental objections.

Reviewer #2: Thank you, that was all for me.

Reviewer #3: The authors addressed the suggestions satisfactorily and the paper has improved in revision. Nice work!

**Have all data underlying the figures and results presented in the manuscript been provided?**

Reviewer #1: Yes

Reviewer #2: Yes

Reviewer #3: Yes

PLOS authors have the option to publish the peer review history of their article (what does this mean?). If published, this will include your full peer review and any attached files.

Reviewer #1: No

Reviewer #2: **Yes: **Maxym Myroshnychenko

Reviewer #3: **Yes: **Stefano Panzeri

---

## [Editor Report · Acceptance letter]

26 May 2021

PCOMPBIOL-D-20-01987R1 

Embedding optimization reveals long-lasting history dependence in neural spiking activity

Dear Dr Rudelt,

I am pleased to inform you that your manuscript has been formally accepted for publication in PLOS Computational Biology. Your manuscript is now with our production department and you will be notified of the publication date in due course.

With kind regards,

Katalin Szabo
